# Learning Time-Varying Multi-Region Brain Communications via Scalable Markovian Gaussian Processes

**Weihan Li** [1]  **Yule Wang** [1]  **Chengrui Li** [1]  **Anqi Wu** [1]

## Abstract

Understanding and constructing brain communications that capture dynamic communications across multiple regions is fundamental to modern system neuroscience, yet current methods struggle to find time-varying region-level communications or scale to large neural datasets with long recording durations. We present a novel framework using Markovian Gaussian Processes to learn brain communications with time-varying temporal delays from multi-region neural recordings, named Adaptive Delay Model (ADM). Our method combines Gaussian Processes with State Space Models and employs parallel scan inference algorithms, enabling efficient scaling to large datasets while identifying concurrent communication patterns that evolve over time. This time-varying approach captures how brain region interactions shift dynamically during cognitive processes. Validated on synthetic and multi-region neural recordings datasets, our approach discovers both the directionality and temporal dynamics of neural communication. This work advances our understanding of distributed neural computation and provides a scalable tool for analyzing dynamic brain networks. Code is available at https://github.com/BRAINML-GT/Adaptive-Delay-Model.

## 1. Introduction

Modern system neuroscience faces a significant challenge in discovering communication patterns that capture dynamic interactions across multiple regions. Understanding these time-varying communications has become increasingly critical with the advent of advanced recording technologies that enable simultaneous measurement of neural activity across numerous brain areas with unprecedented temporal and spatial resolution (Steinmetz et al., 2021; Siegle et al., 2021; Li et al., 2024a; Nejatbakhsh et al., 2024). These large-scale neural recordings necessitate computational methods capable of uncovering and characterizing the dynamic nature of neural communication within the brain.

Latent representations offer a promising approach to building time-varying multi-region communications (Wang et al., 2023; 2024; Zhang et al., 2024). In the brain, communication patterns between brain regions manifest at varying temporal scales: some regions exhibit fast, synchronous interactions with short delays, indicative of strong functional coupling, while others display slower interactions with longer delays, reflecting more indirect relationships. Such communications provide a framework for understanding how these diverse communication patterns evolve over time, capturing both feedforward and feedback pathways that may shift during different cognitive states (Lillicrap et al., 2020; Liu et al., 2025).

Current computational approaches for modeling multi-region communications can be broadly categorized into non-delay models and delay models. Non-delay models, such as mp-srSLDS (Glaser et al., 2020) and MR-SDS (Karniol-Tambour et al.), do not explicitly incorporate temporal delays when capturing latent dynamics across regions. As a result, they can only learn the directional information of communications. Delay models, including DLAG (Gokcen et al., 2022), m-DLAG (Gokcen et al., 2024a), their approximated versions (Gokcen et al., 2024b), and MRM-GP (Li et al., 2024b), introduce mechanisms to learn temporal delays between pairs of communications. Learning delays provides not only directional information but also insights into relative communication speeds, which can help assess the strength of functional coupling between brain regions.

Although these methods have shown potential in capturing inter-region brain communications, delay models, such as DLAG and its variants, do not account for the dynamic nature of multi-region communication. They fail to model time-varying temporal dependencies and communication patterns, which are crucial for understanding neural processing. Non-delay models, such as MR-SDS and mp-srSLDS,

---

[1]School of Computational Science & Engineering, Georgia Institute of Technology, Atlanta, USA. Correspondence to: Anqi Wu <anqiwu@gatech.edu>.

*Proceedings of the $42^{nd}$ International Conference on Machine Learning*, Vancouver, Canada. PMLR 267, 2025. Copyright 2025 by the author(s).

introduce dynamic message flow among regions but don't model delays between communications and assume a single communication subspace, meaning they cannot capture concurrent communications over different subspaces.

MRM-GP is a specific case that integrates a Gaussian Process (GP) with a State Space Model (SSM), making it a delay model with discrete changing of phase delays. MRM-GP has two key limitations: (1) it only supports kernels that are separable across spatial and temporal domains, restricting its applicability in the frequency domain by learning phase delays, whereas temporal delays are more generalizable for neuroscience applications; (2) it assumes that the delays across all communication subspaces across regions change synchronously with hidden state transition. However, this assumption does not reflect the true brain mechanism, where different communication pathways between regions can operate asynchronously.

In terms of computational efficiency, DLAG and mDLAG, which are GP-based methods, incur an $O(T^3)$ computational cost, where $T$ is the number of time samples, making them challenging to scale. Approximated GP and SSM-based methods, such as MRM-GP, MR-SDS, and mp-srSLDS, reduce the cost to $O(T)$. However, they rely on sequential inference, which remains inefficient for large neural datasets with long recording durations. Moreover, the reliance on discrete hidden states, e.g., MRM-GP, introduces inefficiencies during inference.

To address these limitations, we propose the Adaptive Delay Model (ADM). It falls within the family of Markovian Gaussian process-based delay models (Li et al., 2024b), but incorporates a continuous time-varying temporal delay. Thus unlike static communication models commonly used in neuroscience, ADM can accommodate communication patterns with varying temporal characteristics, offering a more flexible and biologically relevant framework.

Markovian Gaussian Process (Markovian GP) models integrate the expressive power of GPs with the computational efficiency of SSMs, facilitating scalable analysis of large-scale neural recordings while discovering multiple evolving communication patterns. However, existing Markovian GP approaches either rely on single-output kernels or require multi-output kernels to be separable across spatial and temporal domains (Solin et al., 2016; Loper et al., 2021; Dowling et al., 2021; 2023; Li et al., 2024b). In this paper, we introduce a novel, universal connection between arbitrary temporally stationary GPs and SSMs, making the framework highly flexible and broadly applicable to various neuroscience problems.

Additionally, we apply an advanced inference method for Markovian GP, leveraging parallel scan algorithms (Blelloch, 1990; Särkkä & García-Fernández, 2020) to signif-

icantly accelerate computation and reducing complexity to $O(\log T)$. This approach enables efficient analysis of long-duration recordings while capturing dynamic communication patterns.

We validate our model using neural recordings from multiple regions of the brain during visual processing tasks (Semedo et al., 2019; Siegle et al., 2021). Our results demonstrate the method's capability to uncover how information flow patterns dynamically change across multi-region networks, offering new insights into the temporal organization of large-scale neural circuits and advancing our understanding of distributed neural computation.

In summary, the key contributions of our work are:

• We establish a universal connection between arbitrary temporally stationary GPs and SSMs, which has broader implications for other domains where computational efficiency is a priority.

• Our model discovers time-varying multi-region communications from latent representations of neural recordings without introducing additional discrete hidden states in the SSM.

• We propose a scalable method for analyzing multi-region communications in large-scale neural data with $O(\log T)$ complexity.

## 2. Method

We begin by demonstrating how a Gaussian Process (GP) with a Factor Analysis (FA) model can be used to capture static brain communication across regions (Section 2.1). Next, we establish a universal connection between a GP and a State Space Model (SSM), referred to as the Markovian Gaussian Process (Section 2.2). Finally, we illustrate how time-varying brain communication can be modeled (Section 2.3).

### 2.1. Gaussian Process Factor Analysis for Brain Communications

The Gaussian Process Factor Analysis (GPFA) for modeling brain communication employs the multi-output Squared Exponential (MOSE) kernel (Gokcen et al., 2022):

$$\mathbf{K}_{ij}(\tau) = \exp\left(-\frac{(\tau + \theta_{ij})^2}{2l^2}\right), \qquad (1)$$

where $i, j$ represent two brain regions, $\tau = t - t'$ is the time difference, $\theta_{ij}$ is the temporal delay between regions $i$ and $j$, and $l$ is the length scale shared across all regions. This MOSE kernel allows the identification of temporal delays that characterize communication dynamics across regions.

Our goal is to learn $MN$ latent variables, $\boldsymbol{x} \in \mathbb{R}^{MN \times T}$,

from neural recordings $\boldsymbol{y} \in \mathbb{R}^{D \times T}$ across $N$ regions. Each region is associated with $M$ latent variables.

A typical assumption for $\boldsymbol{x}$ is to decompose it into two components: across-region variables and within-region variables (Gokcen et al., 2022; Li et al., 2024b). The across-region variables, $\boldsymbol{x}^a \in \mathbb{R}^{m_a N \times T}$, capture shared neural activity that reflects communication between brain regions. These variables exhibit the similar dynamics across regions, differing only in temporal delays. The within-region variables, $\boldsymbol{x}^w \in \mathbb{R}^{m_w N \times T}$, represent neural activity unique to individual regions and are independent of other regions. Together, these components form the latent representation of neural recordings, $\boldsymbol{x} = [\boldsymbol{x}^a, \boldsymbol{x}^w]$, where $m_a + m_w = M$. The relationship between $\boldsymbol{y}$ and $\boldsymbol{x}$ is then modeled using a Factor Analysis (FA) model:

$$\boldsymbol{y} = \mathbf{C}\boldsymbol{x} + d + \epsilon, \qquad (2)$$

where $\mathbf{C} \in \mathbb{R}^{D \times MN}$ is a block-diagonal matrix $\mathbf{C} = \text{diag}\{\mathbf{C}^1, \dots, \mathbf{C}^N\}$, with each $\mathbf{C}^i$ representing the mapping from region $i$'s latent variable to its neural recordings. Additionally, $d \in \mathbb{R}^{D \times 1}$ is a bias term, and $\epsilon \sim \mathcal{N}(0, \mathbf{V})$ represents Gaussian noise with diagonal covariance $\mathbf{V} \in \mathbb{R}^{D \times D}$.

The across-region variables $\boldsymbol{x}^a$ are designed to capture communication patterns among regions. For the $m$-th group of across-region variables, $\boldsymbol{x}_m^a \in \mathbb{R}^{N \times T}$, the activity from each region exhibits spatial correlations with the $N-1$ other regions. These variables are modeled as a Gaussian Process (GP) with the MOSE kernel, which captures the temporal delay characteristics of across-region communication.

On the other hand, the $m$-th group of within-region variables $\boldsymbol{x}_m^w \in \mathbb{R}^{N \times T}$, representing region-specific activity, is modeled independently across regions using a single-output Squared Exponential (SE) kernel:

$$\mathbf{K}^{\text{single}}(\tau) = \exp\left(-\frac{\tau^2}{2l^2}\right). \qquad (3)$$

Additionally, independence is assumed across different groups of both the across-region and within-region variables, with each group $m$ governed by distinct kernel parameters.

By explicitly separating the across-region and within-region latent variables, this framework offers a clear representation of across-region communication and region-specific dynamics, enabling a more interpretable analysis of multi-region neural recordings.

### 2.2. Connect Gaussian Process with State Space Model

We develop a novel universal connection between arbitrary temporally stationary GPs and SSMs, enabling us to efficiently model both across- and within-region dynamics.

**Gaussian Process and State-Space Approximation.** The $m$-th group of across-region variables, $\boldsymbol{x}_m^a \in \mathbb{R}^{N \times T}$ are

modeled as a GP with MOSE kernel:

$$\mathcal{GP}\left(0, \begin{bmatrix} \mathbf{K}(0) & \mathbf{K}(-1) & \dots & \mathbf{K}(-T+1) \\ \mathbf{K}(1) & \mathbf{K}(0) & \dots & \mathbf{K}(-T+2) \\ \vdots & \vdots & \ddots & \vdots \\ \mathbf{K}(T-1) & \mathbf{K}(T-2) & \dots & \mathbf{K}(0) \end{bmatrix}\right), \quad (4)$$

where each $\mathbf{K}(\tau) \in \mathbb{R}^{N \times N}$ is a MOSE kernel in Eq. 1 with an interval $\tau$ over $N$ brain regions. Our goal is to find a state-space approximation of $\boldsymbol{x}_m^a$, which follows a Multi-Order SSM structure:

$$\boldsymbol{x}_{m,t}^a = \sum_{p=1}^{P} \mathbf{A}_p \boldsymbol{x}_{m,t-p}^a + \boldsymbol{q}_t, \quad \boldsymbol{q}_t \sim \mathcal{N}(0, \mathbf{Q}), \quad (5)$$

where $P$ represents the number of orders, $\mathbf{A}_1, \dots, \mathbf{A}_P \in \mathbb{R}^{N \times N}$ are the transition matrices, and $\mathbf{Q} \in \mathbb{R}^{N \times N}$ is the process noise matrix.

**Determining an State Space Model using Kernels.** To estimate transition matrices and measurement using $\boldsymbol{x}_m^a$, we can consider the SSM in Eq. 5 as a regression model (Neumaier & Schneider, 2001):

$$\boldsymbol{x}_{m,t}^a = \mathbf{G}\boldsymbol{v}_t + \boldsymbol{q}_t, \quad \boldsymbol{q}_t \sim \mathcal{N}(0, \mathbf{Q}), \qquad (6)$$

where $\mathbf{G} \in \mathbb{R}^{N \times NP}$ is the regression coefficient and $\boldsymbol{v}_t \in \mathbb{R}^{NP \times 1}$ is the predictor:

$$\begin{aligned} \mathbf{G} &= [\mathbf{A}_P, \mathbf{A}_{P-1}, \dots, \mathbf{A}_1], \\ \boldsymbol{v}_t &= [\boldsymbol{x}_{m,t-P}^{a,\top}, \boldsymbol{x}_{m,t-P+1}^{a,\top}, \dots, \boldsymbol{x}_{m,t-1}^{a,\top}]^\top. \end{aligned} \qquad (7)$$

Our ultimate goal is to use $\mathbf{K}(\tau)$ to represent $\mathbf{G}$ and $\mathbf{Q}$. First, we can represent $\mathbf{G}$ and $\mathbf{Q}$ as functions of $\boldsymbol{x}_m^a$. Concretely, given $T$ samples, $\boldsymbol{x}_{m,1}^a, \dots, \boldsymbol{x}_{m,T}^a$, we define predictor matrix as $\mathbf{V} = [\boldsymbol{v}_{P+1}, \dots \boldsymbol{v}_T]^\top \in \mathbb{R}^{NP \times (T-P)}$ and target observation matrix as $\mathbf{W} = [\boldsymbol{x}_{m,P+1}^a, \dots, \boldsymbol{x}_{m,T}^a] \in \mathbb{R}^{N \times (T-P)}$.

Then, we can represent the regression model in Eq. 6 in the matrix form: $\mathbf{W} = \mathbf{G}\mathbf{V} + \mathbf{R}$, where $\mathbf{R}$ is the residual matrix. By doing so, we can estimate coefficient matrix $\mathbf{G}$ and the process noise matrix $\mathbf{Q}$ by least squares estimation:

$$\mathbf{G} = \mathbf{W}\mathbf{V}^\top (\mathbf{V}\mathbf{V}^\top)^{-1}, \quad \mathbf{Q} = \frac{\mathbf{R}\mathbf{R}^T}{T-P-1}, \qquad (8)$$

where $\mathbf{R} = \mathbf{W} - \mathbf{G}\mathbf{V}$ denotes an estimate of the residual matrix, and its covariance is an estimate of process noise matrix $\mathbf{Q}$. Now, if we can represent $\mathbf{W}\mathbf{V}^\top$ and $\mathbf{V}\mathbf{V}^\top$ with $\mathbf{K}(\tau)$, we will achieve the ultimate goal.

Since each sample $\boldsymbol{x}_{m,t}^a$ in $\mathbf{V}$ and $\mathbf{W}$ is modeled as a sample in the GP in Eq. 4. We can represent $\mathbf{V}\mathbf{V}^\top \in \mathbb{R}^{NP \times NP}$

and $\mathbf{W}\mathbf{V}^\top \in \mathbb{R}^{N \times NP}$ using $\mathbf{K}(\tau)$ as (full derivations see Appendix A):

$$
\mathbf{V}\mathbf{V}^\top \propto \begin{bmatrix} \mathbf{K}(0) & \mathbf{K}(-1) & \dots & \mathbf{K}(-P+1) \\ \mathbf{K}(1) & \mathbf{K}(0) & \dots & \mathbf{K}(-P+2) \\ \vdots & \vdots & \ddots & \vdots \\ \mathbf{K}(P-1) & \mathbf{K}(P-2) & \dots & \mathbf{K}(0) \end{bmatrix},
$$

$$
\mathbf{W}\mathbf{V}^\top \propto \begin{bmatrix} \mathbf{K}(P) & \mathbf{K}(P-1) \dots & \mathbf{K}(1) \end{bmatrix}. \tag{9}
$$

Notably, each $\mathbf{K}(\tau) \in \mathbb{R}^{N \times N}$, where $\tau \in [-P+1, P-1]$, depends only on the number of brain regions $N$ and can be efficiently computed using the stationary kernel function employed in the GP. Furthermore, $\mathbf{K}(\tau)$ can represent any stationary temporal kernel, establishing a universal connection between GPs and SSMs. We apply this universal conversion to various kernels in GP regression task, see Appendix D for details.

**Markovian Across-region Communcations.** Now, the transition matrices and the measurement matrix in Eq. 5 are uniquely determined by the kernel functions of GP by Eq. 8 and Eq. 9. Moreover, we can rewrite the SSM in Eq. 5 to an SSM with a Markovian structure, resulting in a Markovian Gaussian Process (Markovian GP) (Zhao, 2021):

$$
\hat{\boldsymbol{x}}_{m,t}^a = \hat{\mathbf{A}}\hat{\boldsymbol{x}}_{m,t-1}^a + \boldsymbol{q}_t, \quad \boldsymbol{q}_t \sim \mathcal{N}(0, \hat{\mathbf{Q}}),
$$
$$
\boldsymbol{x}_{m,t}^a = \mathbf{H}\hat{\boldsymbol{x}}_{m,t}^a, \tag{10}
$$

where $\mathbf{H} \in \mathbb{R}^{N \times NP}$ denotes a mask matrix, $\hat{\mathbf{A}} \in \mathbb{R}^{NP \times NP}$ is structured as a controllable canonical form (Grewal & Andrews, 2014) and small constants are added to $\hat{\mathbf{Q}} \in \mathbb{R}^{NP \times NP}$ so it matches the shape of $\hat{\mathbf{A}}$:

$$
\hat{\mathbf{A}} = \begin{bmatrix} \mathbf{A}_1 & \mathbf{A}_2 & \dots & \mathbf{A}_{P-1} & \mathbf{A}_P \\ \mathbf{I}_N & 0 & \dots & 0 & 0 \\ 0 & \mathbf{I}_N & \dots & 0 & 0 \\ 0 & 0 & \ddots & 0 & 0 \\ 0 & 0 & \dots & \mathbf{I}_N & 0 \end{bmatrix},
$$
$$
\hat{\mathbf{Q}} = \begin{bmatrix} \mathbf{Q} & 0 \\ 0 & \sigma \mathbf{I}_{N(P-1)} \end{bmatrix}, \quad \mathbf{H} = \begin{bmatrix} \mathbf{I}_N & 0 \end{bmatrix}, \tag{11}
$$

with $\mathbf{I}_N \in \mathbb{R}^{N \times N}$ denoting the identity matrix and $\sigma$ a small constant added for numerical stability. Notably, although we use a Markovian structure to represent a stationary GP, our method still incorporates information from multiple orders.

**Markovian Within-region Neural Activity.** Similarly, the state-space approximation of the $m$-th group of within-region variables, $\boldsymbol{x}_m^w \in \mathbb{R}^{N \times T}$, can be seen as a specific case of across-region variables. In this case, each dimension, $\boldsymbol{x}_{m,n}^w \in \mathbb{R}^{T \times 1}$, is independently modeled as a Markovian GP with a scalar single-output SE kernel (Eq. 3).

## 2.3. Time-Varying Brain Communications

Since the SSM in Eq. 10 follows a discrete structure, we can extend it to incorporate time-varying transition and process noise matrices as follows:

$$
\hat{\boldsymbol{x}}_{m,t}^a = \hat{\mathbf{A}}_t \hat{\boldsymbol{x}}_{m,t-1}^a + \boldsymbol{q}_t, \quad \boldsymbol{q}_t \sim \mathcal{N}(0, \hat{\mathbf{Q}}_t),
$$
$$
\boldsymbol{x}_{m,t}^a = \mathbf{H}\hat{\boldsymbol{x}}_{m,t}^a. \tag{12}
$$

This formulation introduces a time-varying MOSE kernel, where the temporal delay parameter $\theta_{ij,t}$ evolves over time. In other words, at each time step $t$, we construct a Markovian GP (or SSM) as described in Eq. 10, conditioned on the MOSE kernel specific to that time $t$. Additionally, the length scale parameter $l$ is held constant over time, which limits the flexibility of each $\hat{\mathbf{A}}_t$. By sharing $l$ across time points, we constrain the temporal evolution of the delay parameters, promoting smoother dynamics and reducing the risk of misattributing variability in the messages to fluctuations in delays.

This approach enables the model to learn time-varying temporal delays, effectively capturing the dynamics of multi-region brain communications. Compared to modeling time-varying phase delays using hidden discrete states (Li et al., 2024b), our method is more flexible, as it does not assume that each group of across-region communications, $\boldsymbol{x}_m^a$, undergoes simultaneous delay changes during state transitions.

Importantly, this computation can be efficiently performed in vectorized form across all $T$ time steps, ensuring minimal impact on overall computational efficiency. In the FA model, the projection matrix $\mathbf{C} \in \mathbb{R}^{D \times MN}$, the bias term $d \in \mathbb{R}^{D \times 1}$, and the observation Gaussian noise $\epsilon \in \mathbb{R}^{D \times 1}$ remain time-invariant.

## 3. Inference

Now, having established the connection between the $m$-th group of across-region communications and within-region neural activity to the Markovian GP (or SSM), as described in Eq. 12, the next step is to efficiently learn the latent variables and model parameters.

Our model, ADM, formulated as an SSM, offers a significant advantage: it can learn the parameters using either a sequential estimation method with complexity $O(T)$ or a parallel computation method with complexity $O(\log T)$. On modern hardware, the parallel approach is consistently faster due to its efficient utilization of computational resources.

**Parameter Settings** The model parameters, denoted as $\Theta$, include the kernel parameters $\theta_{ij,t}^k$ and $l^k$ from each latent dimension $k$, which define the transition matrix $\hat{\mathbf{A}}_t$ and the process noise covariance matrix $\hat{\mathbf{Q}}_t$ for each across- or within-region latent group. Additionally, the Factor Analysis (FA) parameters include the projection matrix $\mathbf{C}$, the

bias term $d$, and the diagonal matrix $\mathbf{V}$. The model also has a hyperparameter $P$, representing the order of the autoregressive process. To better understand the effect of different $P$ values, we generate samples from our model for various $P$ values, as shown in Appendix C.

**Parallel Scan Kalman EM Algorithm**  Given neural recordings $\boldsymbol{y} \in \mathbb{R}^{D \times T}$, our goal is to estimate the latent brain communications $\boldsymbol{x} \in \mathbb{R}^{MN \times T}$ along with the model parameters $\Theta$. To achieve this, we use the parallel scan-based Kalman Expectation-Maximization (EM) inference algorithm (Särkkä & García-Fernández, 2020), which introduces a parallel scan version of the Kalman Filter and Smoother. Specifically, in the E-step, we apply the parallel Kalman Filter and Smoother to infer the latent variables and expected log-likelihood. In the M-step, we update the kernel parameters using gradient descent and optimize the Factor Analysis (FA) parameters through closed-form linear regression. See Appendix B for details.

The objective of the Kalman Filter is to compute the posterior density $p(\boldsymbol{x}_t | \boldsymbol{y}_{0:t})$, given the neural data up to time step $t$, while the Kalman Smoother computes the posterior density $p(\boldsymbol{x}_t | \boldsymbol{y}_{0:T})$ for all time steps. Traditionally, both filtering and smoothing are computed in $O(T)$ time using sequential updates. However, sequential computation is often inefficient compared to parallel computation, particularly on modern hardware architectures (Chen et al., 2024).

To address this inefficiency, Särkkä & García-Fernández (2020) demonstrates that the sequential updates of the Kalman Filter and Smoother can be reformulated as an associative operator, enabling the use of the parallel scan algorithm (Blelloch, 1990). Consequently, the time complexity and memory cost of our model are given by:

$$\text{Time complexity:} \quad O(MN^3P^3 \log T), \quad (13)$$

$$\text{Memory complexity:} \quad O(MN^2P^2T), \quad (14)$$

where $N$ is the number of regions, $M$ is the group number of across- and within-region latent dynamics, and $P$ is the SSM order in Eq. 5. The cubic cost arises from Eq. 8. However, as we will show in the experimental section, the order parameter $P$ and $N$ can be significantly smaller compared to $T$ while still achieving strong generative and inference performance. Thus, the cubic cost does not pose a major computational bottleneck.

## 4. Experiment

**Datasets.**  We evaluate our model on three datasets.

● **Synthetic Data**: We generate synthetic data that incorporate both across-region communications and within-region neural activities, along with time-varying temporal delays, to simulate dynamic brain communications characterized by both fast and slow features.

● **Two Brain Regions** (Semedo et al., 2019; Zandvakili & Kohn, 2019): Simultaneous spike train recordings from a monkey's primary visual area (V1) and secondary visual cortex (V2), with a 6Hz drifting grating as the external stimulus.

● **Five Brain Regions** (Siegle et al., 2021): Simultaneous spike train recordings from a mouse's primary visual cortex (VISp), rostrolateral area (VISrl), anterolateral area (VISal), posteromedial area (VISpm), and anteromedial area (VISam), with a 4Hz drifting grating as the external stimulus.

**Baselines.**  We compare our model with three methods:

● **DLAG** (Gokcen et al., 2022): A Gaussian Process Factor Analysis model with a MOSE kernel, designed for neural recordings from two brain regions. It can be used to learn both across-region and within-region latent communications.

● **mDLAG** (Gokcen et al., 2024a): An extension of DLAG that supports more than two brain regions with a different inference approach. Unlike DLAG, it assumes all latent variables correspond to across-region communications and does not explicitly model within-region dynamics.

● **MRM-GP** (Li et al., 2024b): An approximation of a Gaussian Process with a Cross-Spectral Mixture (CSM) kernel (Ulrich et al., 2015), formulated as an SSM with $O(T)$ complexity. It is designed to learn frequency-based communications between two brain regions and can capture both across-region and within-region latent dynamics.

**Evaluation.**  We evaluate our model and baseline models by randomly splitting the data into training, validation, and testing sets with a ratio of 0.8, 0.1, and 0.1, respectively. Since all models assume a linear/Gaussian relationship between the latent variables and observed data, we assess their performance by computing the observation test log-likelihood: $\text{LL}(\boldsymbol{x}_{\text{test}}, \boldsymbol{y}_{\text{test}})$, where $\boldsymbol{x}_{\text{test}}$ represents the inferred test latent variables, and $\boldsymbol{y}_{\text{test}}$ denotes the test neural recordings. To mitigate randomness, we report the average test log-likelihood over five different random seeds.

### 4.1. Synthetic Data

In this section, we simulate a common phenomenon in neuroscience where brain region communications are dynamic (Parra & Tobar, 2017). Our goal is to evaluate our model's ability to recover time-varying temporal delays and latent variables.

**Experimental setup.**  We generate 120 independent trials for two brain regions ($N = 2$) with an order of $P = 5$

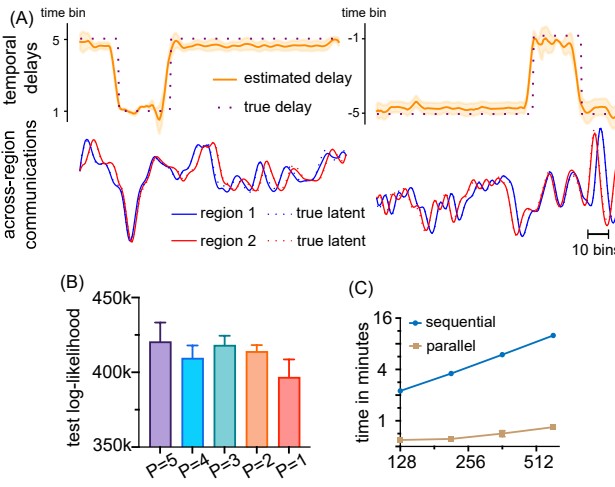

*Figure 1.* Evaluation of the ADM on synthetic brain communication data. (A) Estimated across-region communications, temporal delays, and ground truth over $T$ bins for $P = 5$. The shaded area represents the variance of the estimated delay across five different runs. (B) Test log-likelihood summed over trials and time bins, showing that performance remains stable for different $P$. (C) Comparison of the runtime for sequential and parallel scan-based Kalman EM across different values of $T$, showing that the parallel version is significantly faster, especially as $T$ increases.

and $T = 200$ time bins. Each region contains 50 neurons, with $m_a = 2$ groups of across-region communications and $m_w = 1$ group of within-region variables. For across-region communications, the first group represents forward communication, characterized by a larger positive delay of 5 bins and a smaller positive delay of 1 bin during time bins 30 to 70. The second group represents feedback communication, with a larger negative delay of -5 bins and a smaller negative delay of -1 bin during time bins 130 to 170. For across-region and within-region dynamics, the length scales are set to $l = 5$ and $l = 2.5$, respectively.

Note that large delays in this context are intended to represent slow communication in the brain, whereas an extremely large delay would imply an absence of communication between regions. A small delay indicates fast communication. Therefore, our data simulation is designed to reflect a scenario where region A initially has minimal effective communication with region B (delay of 5), then suddenly transmits a signal (delay of 1), followed by another period of ineffective communication (delay of 5). A delay of -5 represents communication in the opposite direction. During fitting, we set $m_a = 2$, $m_w = 1$, and $P = 5$.

**Results.** Figure 1(A) presents the estimated and truth across-region communications, temporal delays, and ground truth over $T$ bins for $P = 5$. For the estimated delay, the shaded area represents the variance across five different runs.

Our model effectively captures time-varying communications for both latent dynamics and delay. See Appendix E for within-region neural activities.

Figure 1(B) shows the test log-likelihood summed over trials and time bins. The results indicate that performance remains relatively stable for $P \in [2, 5]$, except for $P = 1$, which yields the lowest performance.

Figure 1(C) compares the time costs of the sequential and parallel scan-based Kalman EM algorithms with GPU parallelization. We generate synthetic data with up to $T = 600$ time bins. The results demonstrate that the parallel version is significantly faster than the sequential update.

Finally, Appendix G presents a more complex case with five-region synthetic dataset and evaluates the model's performance as a function of the number of regions, latent variables, and data length.

### 4.2. Two Brain Regions

In this section, we investigate the interactions between the mouse's primary visual area (V1) and secondary visual cortex (V2) in response to a 6Hz drifting grating. Additionally, we compare our model's performance and inference time with MRM-GP and DLAG.

**Experimental setup.** We use smoothed multi-region spike trains from session 106r001p26 with an orientation of $0°$. This dataset consists of 400 trials, each containing 64 time bins (20 ms per bin), with 72 V1 neurons and 22 V2 neurons. The monkey begins receiving the visual stimulus (drifting gratings) at the first time bin, and the stimulus persists throughout all 64 time bins. The number of across-region and within-region latent dynamics follows previous works (Gokcen et al., 2022; Li et al., 2024b), where $m_a = 2$ and $m_w = 2$. The order $P = 4$ is selected based on performance evaluation on the validation dataset.

**Results.** Figure 2(A) shows the estimated across-region communications and time-varying delays from the test dataset (ten trials shown; within-region dynamics are provided in Appendix E). The first group of communications shows a shift from slower feedback (with larger absolute delays) to faster feedback (with smaller absolute delays) starting around 200 ms after stimulus onset. In contrast, the second group exhibits a periodic pattern driven by the external drifting grating stimulus, along with a change in communication direction immediately following stimulus onset. The time-varying delays suggest the following interpretation: shortly after stimulus presentation, V2 generates a strong feedback signal from V2 to V1, potentially reflecting the emergence of surprise or prediction error (Rao & Ballard, 1999). As time progresses, both regions transition into more synchronized oscillations. Our findings on V1-V2

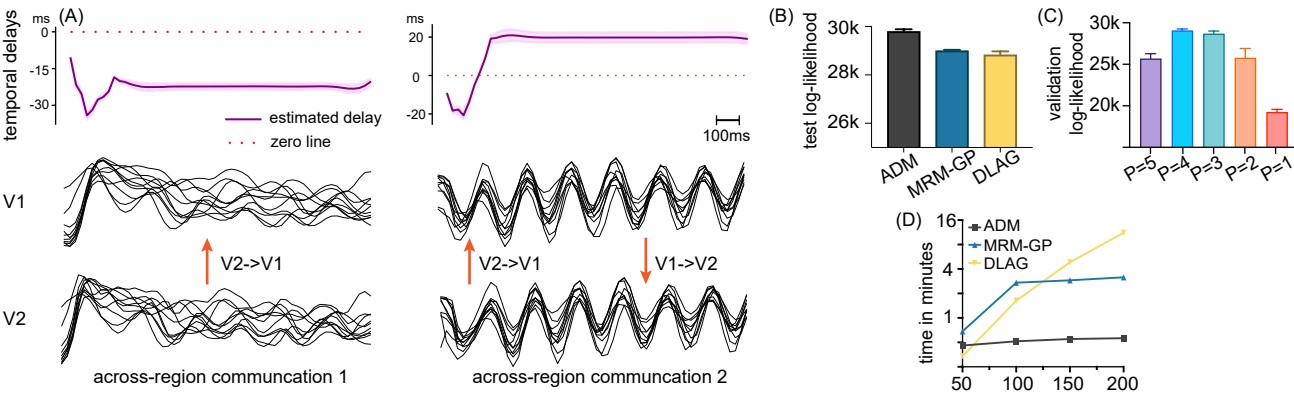

*Figure 2.* Evaluation of the ADM on spike trains from V1 and V2. (A) Estimated across-region communications and time-varying temporal delays from the test dataset (ten trials are shown), showing a time-varying feedback and forward communications between V1 and V2. (B) Test log-likelihood comparison, where ADM outperforms MRM-GP and DLAG by capturing continuously time-varying temporal delays. (C) Validation log-likelihood across different $P$ values. (D) Computational time comparison across different length of $T$, demonstrating ADM's efficiency through parallel computation, outperforming both MRM-GP and DLAG.

interactions are similar to previous studies (Gokcen et al., 2022; Li et al., 2024b; Gokcen et al., 2024a).

Figure 2(B) compares the test log-likelihood summed over trials and time bins, showing that our model, ADM, outperforms MRM-GP and DLAG under the same $m_a$ and $m_w$ settings. This improvement is attributed to ADM's ability to capture continuously time-varying temporal delays. Figure 2(C) presents the validation log-likelihood across different $P$ values, with $P = 4$ achieving the highest value. Combined with the insights from Figures 2(A–B), this suggests that a small $P$ value can effectively estimate model parameters and latent variables.

Figure 2(D) compares the computational time of our model with MRM-GP and DLAG on spike trains of varying $T$, obtained by concatenating trials. The use of parallel computation significantly improves efficiency, outperforming the linear model (MRM-GP) and the cubic model (DLAG).

### 4.3. Five Brain Regions

In this section, we scale up our model to a larger neural recording spanning five regions with increased time resolution. Our objective is to investigate across-region communications and identify the time-varying meso-scale brain network, defined as the dynamic network spanning sAppendixub-brain regions, e.g., regions in visual cortex.

**Experimental setup.** We use smoothed multi-region spike train data from the Visual Coding – Neuropixels project by the Allen Institute (Siegle et al., 2021), specifically from session 750749662. This dataset includes spike trains recorded from VISp, VISrl, VISal, VISpm, and VISam—sub-areas of the mouse visual cortex. It consists of 120 trials, $T = 200$

time bins (each 10 ms), and a total of 202 neurons, with external visual stimuli comprising 4 Hz drifting gratings. Following the approach in (Gokcen et al., 2022), we first apply Factor Analysis to estimate the total number of across-region and within-region latent dynamics, determining $M = 4$ (see Appendix F for details). We then conduct a grid search with 5-fold cross-validation to refine the number of across-region and within-region latent dynamics and the model order $P$.

**Results.** Figure 3(A) presents the ten estimated pairwise temporal delays from one group of across-region communications. See Appendix E for the estimated latent dynamics. Our results reveal consistent forward communication from VISp to downstream visual areas, such as VISrl, VISal, and VISpm, aligning with the known anatomical hierarchy of the mouse visual cortex (Siegle et al., 2021). Additionally, these forward communications exhibit time-varying dynamics. For instance, communication between VISp and VISrl transitions from slow to fast, indicating an enhanced interaction that gradually becomes more synchronous following the initial surprise response to the visual stimulus onset. In contrast, the communication between VISp and VISal shifts from fast to slow, suggesting inhibition induced by the external stimulus. Furthermore, our findings indicate that all communications involving VISam are feedback signals. This is expected, as VISam is positioned at the end of the anatomical hierarchy of the mouse visual system, consistent with the anatomical hierarchy scores reported in (Siegle et al., 2021).

Figure 3(B) depicts the meso-scale brain network corresponding to the across-region communications presented in Figure 3(A). Each node represents a region in the visual system, while directed edges indicate the directional

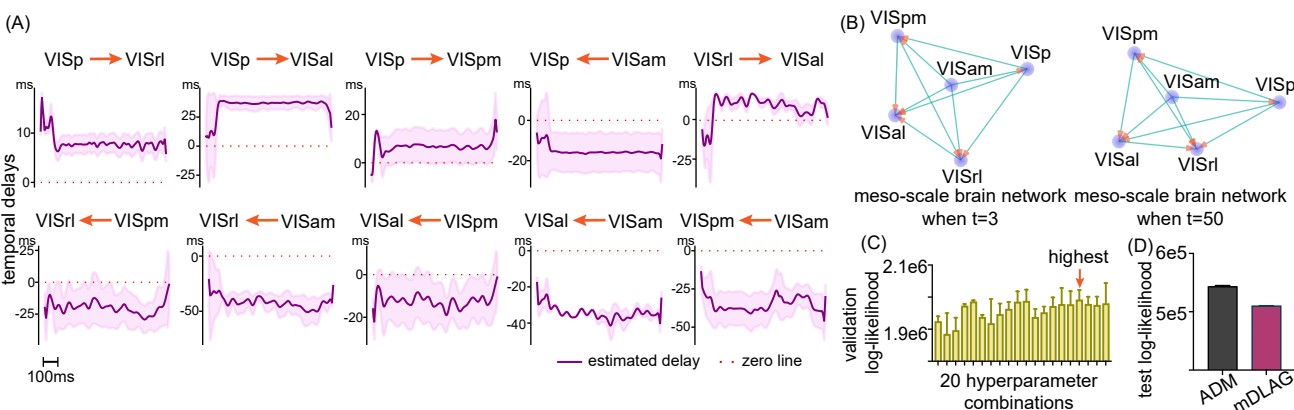

*Figure 3.* Evaluation of the ADM on spike trains from five visual brain regions. (A) Estimated pairwise temporal delays from one group of across-region communications, showing time-varying dynamics and alignment with the anatomical hierarchy of the mouse visual cortex. (B) Meso-scale brain networks at $t = 3$ and $t = 50$ time bins, derived from the communications in (A), illustrating changes in communication speed and direction. (C) Cross-validation results for hyperparameter selection, with $m_a = 3$, $m_w = 1$, and $P = 5$ achieving the highest validation log-likelihood. (D) Test log-likelihood comparison between ADM and mDLAG, demonstrating ADM's superior fit due to its ability to model time-varying communications.

communications. The length of each edge is determined by the estimated delays, reflecting the speed of communication. The figure presents two meso-scale brain networks at $t = 3$ and $t = 50$ time bins. The primary differences between these networks include changes in the speed of certain forward communications and a direction change in the communication direction between VISrl and VISal, suggesting the emergence of stimulus presentation.

Figure 3(C) presents the cross-validation results for twenty hyperparameter combinations. We first determine $M = 4$ using Factor Analysis (see Appendix F for details), then conduct a grid search over all combinations of $m_a \in [0, 4]$, $m_w \in [0, 4]$, and $P \in [2, 5]$. The highest validation log-likelihood is achieved with $m_a = 3$, $m_w = 1$, and $P = 5$.

Figure 3(D) compares the test log-likelihood, summed over trials and time bins, between our model (ADM) and mD-LAG with $m_a = 3$ latent communication channels, where mDLAG is an extension of DLAG that supports more than two brain regions using variational inference. The results indicate that ADM provides a better fit to the data, attributed to its ability to model time-varying communications. We do not compare MRM-GP and DLAG since they are limited to two brain regions. Additionally, we skip a time cost comparison with mDLAG because it is implemented only in MATLAB, which is significantly slower than our GPU-optimized implementation.

## 5. Discussion

**Summary.** Our findings highlight the importance of modeling time-varying multi-region neural communications and

demonstrate that the Adaptive Delay Model (ADM) effectively captures these dynamics while maintaining computational efficiency. Existing methods for studying across-region neural interactions can be broadly categorized into non-delay models and delay models. While non-delay models provide directional communication patterns, they fail to capture temporal delays, limiting their ability to infer the communication speed. Conversely, delay models, such as DLAG and MRM-GP, introduce delay estimation but assume static or discretely changing delays, which do not reflect the continuously evolving nature of brain communications. Our results show that ADM overcomes these limitations by incorporating a flexible, time-varying delay mechanism, enabling a more biologically relevant representation of neural interactions.

**Neuroscience Implications.** Our results from large-scale neural recordings show that across-region communication delays are not static but change over time. Notably, we observe transitions from slow feedback and forward communication to fast forward interactions in both datasets (Section 4.2 and Section 4.3), aligning with adaptive sensory processing in the visual cortex. These findings show the importance of time-varying models in capturing the dynamic nature of brain communications.

**Computational Advancements.** Beyond its neuroscientific implications, our model contributes to the broader field of computational modeling by bridging temporally stationary GPs with SSMs. Traditional GP-SSM connections often rely on separability assumptions in spatial and temporal kernels, limiting their flexibility. Our proposed universal

connection between arbitrary temporally stationary GPs and SSMs removes this restriction. Furthermore, by leveraging parallel scan algorithms, ADM achieves an impressive computational complexity of $O(\log T)$, significantly improving scalability compared to existing methods.

**Limitations and Future Directions.** Our model has a cubic time complexity with respect to the number of brain regions $N$ and the SSM order $P$. Although these values are typically much smaller than $T$, they can still become computational bottlenecks for specific cases. A potential solution may involve leveraging frequency domain techniques. Parnichkun et al. (2024) proposed a state-free SSM with a controllable canonical transition matrix, similar to ours in Eq. 11, and utilized the Fast Fourier Transform to achieve linear scaling in latent size. Similarly, Gokcen et al. (2024b) approximated the GP kernel in the frequency domain to reduce the computational cost to linear in latent size.

## Acknowledgement

This work is supported by National Institutes of Health BRAIN initiative (1U01NS131810).

## Impact Statement

Our work presents the Adaptive Delay Model (ADM), a scalable and biologically relevant framework for uncovering time-varying multi-region neural communications. By bridging Gaussian Processes (GPs) with State Space Models (SSMs), ADM enhances our ability to analyze large-scale neural recordings, offering insights into sensory processing, cognitive flexibility, and neural disorders. Beyond neuroscience, its adaptability to fields such as robotics, autonomous systems, and signal processing broadens its societal impact. Ethically, as machine learning models increasingly shape neuroscience research, it is crucial to ensure their responsible application, avoiding overinterpretation of inferred neural dynamics and considering the broader implications for brain-computer interfaces and neurotechnology.

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

# A. Derivation for $\mathbf{V}\mathbf{V}^\top$, $\mathbf{W}\mathbf{V}^\top$, and $\mathbf{W}\mathbf{W}^\top$

Let's inspect $\mathbf{V}\mathbf{V}^\top \in \mathbb{R}^{NP \times NP}$ first. To simplify the notation, we use $\boldsymbol{x}$ to represent $\boldsymbol{x}_m^a$ in Eq. 7. We have:

$$
\mathbf{V}\mathbf{V}^\top = \begin{bmatrix} \boldsymbol{x}_1 & \boldsymbol{x}_2 & \dots & \boldsymbol{x}_{T-P} \\ \boldsymbol{x}_2 & \boldsymbol{x}_3 & \dots & \boldsymbol{x}_{T-P+1} \\ \vdots & \vdots & \ddots & \vdots \\ \boldsymbol{x}_P & \boldsymbol{x}_{P+1} & \dots & \boldsymbol{x}_{T-1} \end{bmatrix} \begin{bmatrix} \boldsymbol{x}_1^\top & \boldsymbol{x}_2^\top & \dots & \boldsymbol{x}_P^\top \\ \boldsymbol{x}_2^\top & \boldsymbol{x}_3^\top & \dots & \boldsymbol{x}_{P+1}^\top \\ \vdots & \vdots & \ddots & \vdots \\ \boldsymbol{x}_{T-P}^\top & \boldsymbol{x}_{T-P+1}^\top & \dots & \boldsymbol{x}_{T-1}^\top \end{bmatrix}
$$
$$
= \begin{bmatrix} \boldsymbol{x}_1\boldsymbol{x}_1^\top + \cdots + \boldsymbol{x}_{T-P}\boldsymbol{x}_{T-P}^\top & \dots & \boldsymbol{x}_1\boldsymbol{x}_P^\top + \boldsymbol{x}_2\boldsymbol{x}_{P+1}^\top + \cdots + \boldsymbol{x}_{T-P}\boldsymbol{x}_{T-1}^\top \\ \vdots & \ddots & \vdots \\ \boldsymbol{x}_P\boldsymbol{x}_1^\top + \cdots + \boldsymbol{x}_{T-1}\boldsymbol{x}_{T-P}^\top & \dots & \boldsymbol{x}_P\boldsymbol{x}_P^\top + \boldsymbol{x}_{P+1}\boldsymbol{x}_{P+1}^\top + \cdots + \boldsymbol{x}_{T-1}\boldsymbol{x}_{T-1}^\top \end{bmatrix},
\tag{15}
$$

where the first element $\boldsymbol{x}_1\boldsymbol{x}_1^\top \in \mathbb{R}^{N \times N}$ represents the auto-covariance of $\boldsymbol{x}_1$, which is essentially the kernel $\mathbf{K}(0)$ (Eq. 4). In other words, since $\boldsymbol{x}$ is modeled as a stationary GP, the elements $\boldsymbol{x}_1\boldsymbol{x}_1^\top, \dots, \boldsymbol{x}_{T-1}\boldsymbol{x}_{T-1}^\top$ are all equivalent and correspond to the diagonal elements $\mathbf{K}(0)$ in Eq. 4. Similarly, the elements $\boldsymbol{x}_P\boldsymbol{x}_1^\top, \dots, \boldsymbol{x}_{T-1}\boldsymbol{x}_{T-P}^\top$ represent cross-covariances with time interval $P - 1$, which correspond to the off-diagonal elements $\mathbf{K}(P - 1)$. Therefore, we can further write Eq. 15 as:

$$
\mathbf{V}\mathbf{V}^\top = \begin{bmatrix} \mathbf{K}(0) + \cdots + \mathbf{K}(0) & \dots & \mathbf{K}(-P+1) + \cdots + \mathbf{K}(-P+1) \\ \vdots & \ddots & \vdots \\ \mathbf{K}(P-1) + \cdots + \mathbf{K}(P-1) & \dots & \mathbf{K}(0) + \cdots + \mathbf{K}(0) \end{bmatrix}
$$
$$
\propto \begin{bmatrix} \mathbf{K}(0) & \mathbf{K}(-1) & \dots & \mathbf{K}(-P+1) \\ \mathbf{K}(1) & \mathbf{K}(0) & \dots & \mathbf{K}(-P+2) \\ \vdots & \vdots & \ddots & \vdots \\ \mathbf{K}(P-1) & \mathbf{K}(P-2) & \dots & \mathbf{K}(0) \end{bmatrix}.
\tag{16}
$$

Following the same way, we can also represent $\mathbf{W}\mathbf{V}^\top \in \mathbb{R}^{N \times NP}$ and $\mathbf{W}\mathbf{W}^\top \in \mathbb{R}^{N \times N}$ using $\mathbf{K}$:

$$
\begin{aligned}
\mathbf{W}\mathbf{V}^\top &\propto \begin{bmatrix} \mathbf{K}(P) & \mathbf{K}(P-1) & \dots & \mathbf{K}(1) \end{bmatrix}, \\
\mathbf{W}\mathbf{W}^\top &\propto \mathbf{K}(0).
\end{aligned}
\tag{17}
$$

If the computation of $\mathbf{G}$ in Eq.8 leads to numerical issues because $\mathbf{V}\mathbf{V}^\top$ has singular values that are nearly zero, a more numerically stable approach is to rewrite $\mathbf{V}\mathbf{V}^\top$ by Cholesky factorization:

$$
\mathbf{D} = \begin{bmatrix} \mathbf{V}\mathbf{V}^\top & \mathbf{V}\mathbf{W}^\top \\ \mathbf{W}\mathbf{V}^\top & \mathbf{W}\mathbf{W}^\top \end{bmatrix} = \mathbf{L}\mathbf{L}^\top, \quad \mathbf{L} = \begin{bmatrix} \mathbf{L}_1 & 0 \\ \mathbf{L}_2 & \mathbf{L}_3 \end{bmatrix}, \quad \mathbf{V}\mathbf{V}^\top = \mathbf{L}_1\mathbf{L}_1^\top,
\tag{18}
$$

where $\mathbf{D} \in \mathbb{R}^{N(P+1) \times N(P+1)}$, $\mathbf{W}\mathbf{V}^\top = \mathbf{L}_2\mathbf{L}_1^\top$, $\mathbf{W}\mathbf{W}^\top \propto \mathbf{K}(0)$, and $\mathbf{L}_1 \in \mathbb{R}^{NP \times NP}$, $\mathbf{L}_2 \in \mathbb{R}^{N \times NP}$, $\mathbf{L}_3 \in \mathbb{R}^{N \times N}$ are the sub-matrices of $\mathbf{L}$. In practice, Eq. 18 factorizes $\mathbf{D} + \delta\mathbf{I}$ with a small postive number $\delta$ to ensure the positive definite of $\mathbf{D}$. Then, the estimation for $\mathbf{G}$ can be cast in the form of $\mathbf{L}$ and the measurement matrix $\mathbf{Q}$ is the residual covariance of residual $\mathbf{R}$:

$$
\begin{aligned}
\hat{\mathbf{G}} &= \mathbf{W}\mathbf{V}^\top (\mathbf{V}\mathbf{V}^\top)^{-1} = \mathbf{L}_2\mathbf{L}_1^{-1}, \\
\hat{\mathbf{Q}} &= \frac{(\mathbf{W} - \hat{\mathbf{G}}\mathbf{V})(\mathbf{W} - \hat{\mathbf{G}}\mathbf{V})^\top}{T - P - 1} = \frac{\mathbf{L}_3\mathbf{L}_3^\top}{T - P - 1}.
\end{aligned}
\tag{19}
$$

## B. Details for Kalman EM

We consider the linear–Gaussian state–space model

$$\boldsymbol{x}_t = \mathbf{A}_t \boldsymbol{x}_{t-1} + \boldsymbol{w}_t, \qquad \boldsymbol{w}_t \sim \mathcal{N}(0, \mathbf{Q}_t), \qquad \boldsymbol{y}_t = \mathbf{H}\boldsymbol{x}_t + \boldsymbol{v}_t, \qquad \boldsymbol{v}_t \sim \mathcal{N}(0, \mathbf{V}),$$

where the transition matrix $\mathbf{A}_t$ and process-noise covariance $\mathbf{Q}_t$ change with time. Let $\Theta = \{\mathbf{A}_{1:T}, \mathbf{Q}_{1:T}, \mathbf{H}, \mathbf{V}\}$ denote the full parameter set.

### E–step

With parameters fixed at $\Theta^k$, the expected complete-data log-likelihood is

$$Q(\Theta \mid \Theta^k) = \mathbb{E}_{\mathbf{x}\mid\mathbf{y},\Theta^k}\big[\log p(\mathbf{x}, \mathbf{y} \mid \Theta)\big]$$

$$\propto -\frac{1}{2}\sum_{t=1}^{T}\Big(\boldsymbol{y}_t^\top \mathbf{V}^{-1}\boldsymbol{y}_t - 2\boldsymbol{y}_t^\top \mathbf{V}^{-1}\mathbf{H}\,\widetilde{\mathbf{x}}_t + \mathrm{Tr}\big(\mathbf{H}^\top \mathbf{V}^{-1}\mathbf{H}\mathbf{V}_t\big)\Big)$$

$$-\frac{1}{2}\sum_{t=1}^{T}\Big(\mathrm{Tr}\big(\mathbf{Q}_t^{-1}\mathbf{C}_t\big) - 2\,\mathrm{Tr}\big(\mathbf{Q}_t^{-1}\mathbf{A}_t\mathbf{C}_{t,t-1}^\top\big) + \mathrm{Tr}\big(\mathbf{A}_t^\top \mathbf{Q}_t^{-1}\mathbf{A}_t\mathbf{C}_{t-1}\big)\Big)$$

$$-\frac{T}{2}\log|\mathbf{V}| - \frac{1}{2}\sum_{t=1}^{T}\log|\mathbf{Q}_t|. \tag{20}$$

Here

$$\widetilde{\mathbf{x}}_t = \mathbb{E}[\boldsymbol{x}_t \mid \mathbf{y}, \Theta^k], \quad \mathbf{C}_t = \mathbb{E}[\boldsymbol{x}_t\boldsymbol{x}_t^\top \mid \mathbf{y}, \Theta^k], \quad \mathbf{C}_{t,t-1} = \mathbb{E}[\boldsymbol{x}_t\boldsymbol{x}_{t-1}^\top \mid \mathbf{y}, \Theta^k],$$

which are obtained with a Kalman filter followed by a Rauch–Tung–Striebel smoother run with the time-varying parameters (Boots, 2009). The individual expectations that appear in (20) expand to

$$\mathbb{E}[\boldsymbol{x}_t^\top \mathbf{H}^\top \mathbf{V}^{-1}\mathbf{H}\boldsymbol{x}_t] = \widetilde{\mathbf{x}}_t^\top \mathbf{H}^\top \mathbf{V}^{-1}\mathbf{H}\widetilde{\mathbf{x}}_t + \mathrm{Tr}\big(\mathbf{H}^\top \mathbf{V}^{-1}\mathbf{H}\mathbf{C}_t\big), \tag{21}$$

$$\mathbb{E}[x_t^\top \mathbf{Q}_t^{-1}\boldsymbol{x}_t] = \widetilde{\mathbf{x}}_t^\top \mathbf{Q}_t^{-1}\widetilde{\mathbf{x}}_t + \mathrm{Tr}\big(\mathbf{Q}_t^{-1}\mathbf{C}_t\big), \tag{22}$$

$$\mathbb{E}[\boldsymbol{x}_t^\top \mathbf{Q}_t^{-1}\mathbf{A}_t x_{t-1}] = \widetilde{\mathbf{x}}_t^\top \mathbf{Q}_t^{-1}\mathbf{A}_t\widetilde{\mathbf{x}}_{t-1} + \mathrm{Tr}\big(\mathbf{Q}_t^{-1}\mathbf{A}_t\mathbf{C}_{t,t-1}^\top\big), \tag{23}$$

$$\mathbb{E}[\boldsymbol{x}_{t-1}^\top \mathbf{A}_t^\top \mathbf{Q}_t^{-1}\mathbf{A}_t x_{t-1}] = \widetilde{\mathbf{x}}_{t-1}^\top \mathbf{A}_t^\top \mathbf{Q}_t^{-1}\mathbf{A}_t\widetilde{\mathbf{x}}_{t-1} + \mathrm{Tr}\big(\mathbf{A}_t^\top \mathbf{Q}_t^{-1}\mathbf{A}_t\mathbf{C}_{t-1}\big). \tag{24}$$

### M–step

The M–step maximises (20) with respect to $\Theta$,

$$\Theta^{k+1} = \arg\max_{\Theta} Q(\Theta \mid \Theta^k),$$

where the paramerer sets $\Theta$ is updated either via gradient descent or in closed form, depending on the specific component.

## C. Generation Samples

To better understand the effect of different $P$ values, we generate samples with $T = 200$ time bins from our model using various $P$ values. Figure 4 shows that when $P$ is very small (e.g., $P = 1$), the generated samples appear unsmooth. However, for $P \geq 2$, the generated samples exhibit no noticeable visual differences.

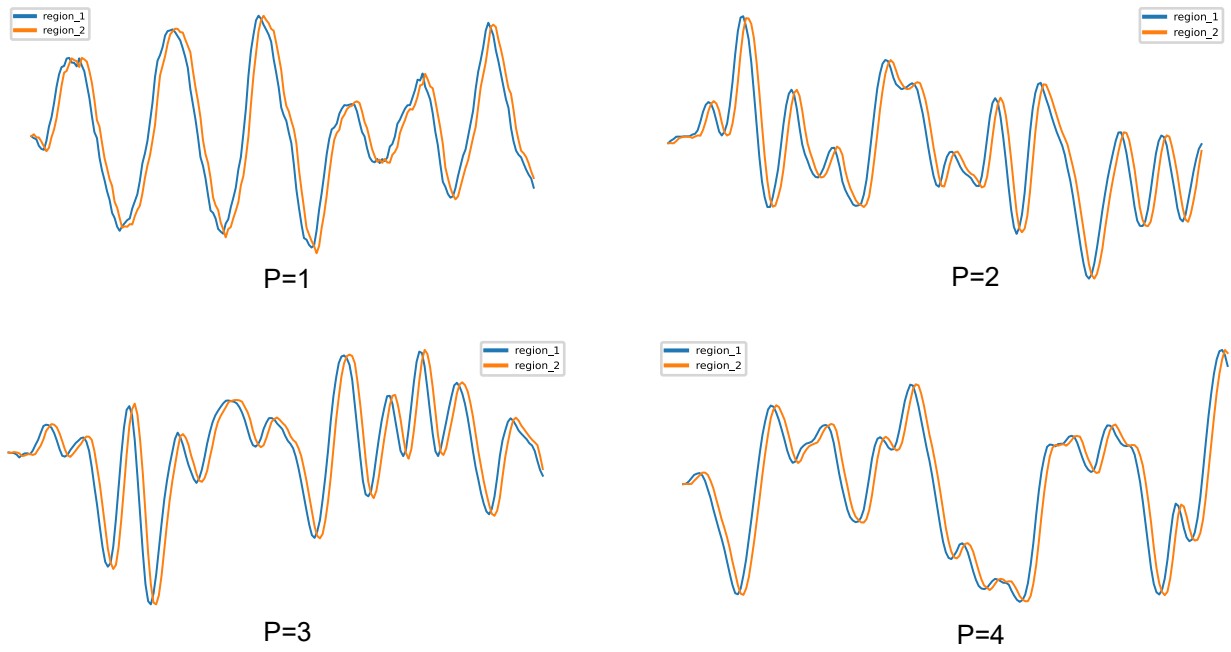

*Figure 4.* Generated samples from our model with MOSE kernel when $P = 1, 2, 3, 4$.

*Table 1.* MSE for GP regression with single-output kernels.

| Reg-MSE / $10^{-1}$ | Exp | Matern $3/2$ | SE | RQ | SM |
|---|---|---|---|---|---|
| GP | $5.7 \pm 0.1$ | $5.9 \pm 0.2$ | $3.1 \pm 0.1$ | $3.0 \pm 0.1$ | $3.0 \pm 0.2$ |
| SSM-Approx | $5.9 \pm 0.1$ | $6.2 \pm 0.1$ | $3.3 \pm 0.1$ | $3.4 \pm 0.1$ | $3.3 \pm 0.2$ |

*Table 2.* MSE for GP regression with multi-output kernels.

| Reg-MSE / $10^{-1}$ | MOSE | MOSM | CSM | LMC |
|---|---|---|---|---|
| GP | $7.4 \pm 0.02$ | $7.5 \pm 0.02$ | $8.2 \pm 0.05$ | $0.66 \pm 0.02$ |
| SSM-Approx | $7.6 \pm 0.04$ | $7.9 \pm 0.08$ | $7.7 \pm 0.09$ | $0.72 \pm 0.02$ |

## D. Gaussian Process Regression

To verify the universal connection between arbitrary temporally stationary Gaussian Processes (GPs) and State Space Models (SSMs), we compare GP regression performance using our SSM approximation and the standard GP. We generate samples of 300 points from a GP with added Gaussian noise as regression data. The samples are then randomly split into training $(t_{\text{train}}, y_{\text{train}})$ and testing $(t_{\text{test}}, y_{\text{test}})$ sets, with 60% used for training and 40% for testing.

The kernels we evaluated are:

- **Exponential (Exp)**: Single-output with $\mathbf{K}(t, t') = \sigma^2 \exp\left(-\frac{|t-t'|}{l}\right)$.
- **Matern 3/2 (Matern)**: Single-output with $\mathbf{K}(t, t') = \sigma^2 \left(1 + \frac{\sqrt{3}|t-t'|}{l}\right) \exp\left(-\frac{\sqrt{3}|t-t'|}{l}\right)$.
- **Squared Exponential (SE)**: Single-output with $\mathbf{K}(t, t') = \sigma^2 \exp\left(-\frac{(t-t')^2}{2l^2}\right)$.
- **Rational Quadratic (RQ)**: Single-output with $\mathbf{K}(t, t') = \sigma^2 \left(1 + \frac{(t-t')^2}{2\alpha l^2}\right)^{-\alpha}$.
- **Spectral Mixture (SM)** (Wilson & Adams, 2013): Single-output with
$\mathbf{K}(t, t') = \sum_{q=1}^{Q} \sigma_q^2 \exp\left(-\frac{(t-t')^2}{2l_q^2}\right) \cos\left(\omega_q(t - t')\right)$.
- **Multi-Output Squared Exponential (MOSE)** (Gokcen et al., 2022): Multi-output with $\mathbf{K}_{ij}(t, t') = \sigma_{ij}^2 \exp\left(-\frac{(t-t'+\delta_{ij})^2}{2l_{ij}^2}\right)$.
- **Multi-Output Spectral Mixture (MOSM)** (Parra & Tobar, 2017): Multi-output with
$\mathbf{K}_{ij}(t, t') = \sum_{q=1}^{Q} \sigma_{ij,q}^2 \exp\left(-\frac{(t-t'+\delta_{ij,q})^2}{2l_{ij,q}^2}\right) \cos\left(\omega_{ij,q}(t - t') + \phi_{ij,q}\right)$.
- **Cross-Spectral Mixture (CSM)** (Ulrich et al., 2015): Multi-output with
$\mathbf{K}_{ij}(t, t') = \sum_{q=1}^{Q} \sum_{r=1}^{R} \sigma_{i,q}^r \sigma_{j,q}^r \exp\left(-\frac{(t-t')^2}{2l_{ij,q}^2}\right) \cos\left(\omega_{ij,q}(t - t') + \phi_{ij,q}^r\right)$.
- **Linear Model of Coregionalization (LMC)**: Multi-output with $\mathbf{K}(t, t') = \sum_{q=1}^{Q} B_q \otimes k_q(t, t')$, where $B_q$ is a coregionalization matrix and $k_q(t, t')$ is a single-output kernel.

The number of orders $P$ for each kernels are as follows:

- **Exponential (Exp)**: $P = 1$.
- **Matern 3/2 (Matern)**: $P = 2$.
- **Squared Exponential (SE)**: $P = 2$.
- **Rational Quadratic (RQ)**: $P = 4$.
- **Spectral Mixture (SM)**: $P = 2$.
- **Multi-Output Squared Exponential (MOSE)**: $P = 2$.
- **Multi-Output Spectral Mixture (MOSM)**: $P = 2$.
- **Cross-Spectral Mixture (CSM)**: $P = 4$.

• **Linear Model of Coregionalization (LMC)**: When $k_q(t, t')$ is SE kernel, $P = 2$.

The results are shown in Table 1 and Table 2, where our SSM approximation demonstrates regression performance comparable to GP in terms of MSE.

# E. Additional Across- and Within-Region Latent Variables

Figure 5 presents the within-region neural activity for both synthetic data and V1-V2 neural spike trains.

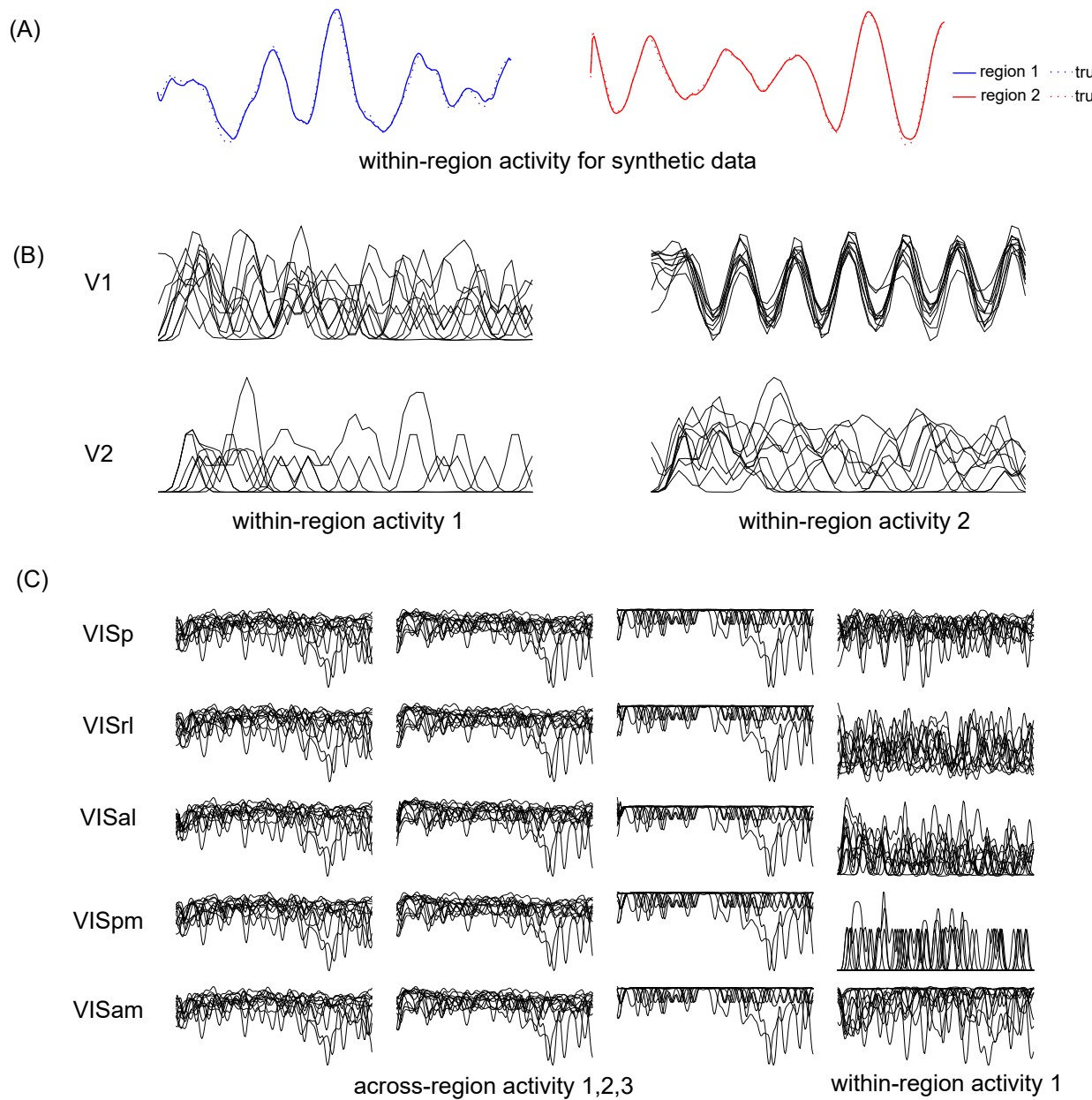

*Figure 5.* (A) Within-region neural activity for synthetic data. (B) Within-region neural activity for V1-V2 neural spike trains (ten trials are shown). (C) The across- and within-region latent variables for the five-region dataset (ten trials are shown).

## F. Factor Analysis for Five Regions Spike Trains

Figure 6 presents the Factor Analysis results for neural spike trains from five regions, and we select the latent size to be the largest optimal latent size across five regions, which is 4.

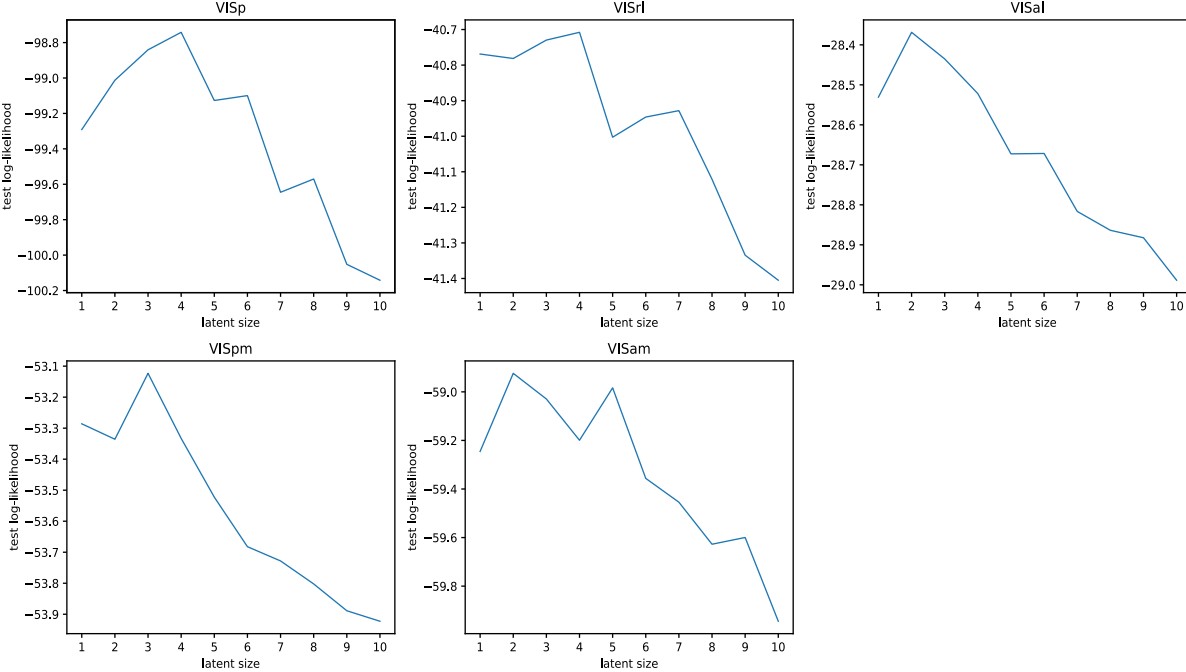

*Figure 6.* The figure presents Factor Analysis results for neural spike trains from five regions. The optimal latent size, determined as the maximum across all regions, is selected to be 4, yielding the highest test log-likelihood.

# G. More Synthetic Data Experiments

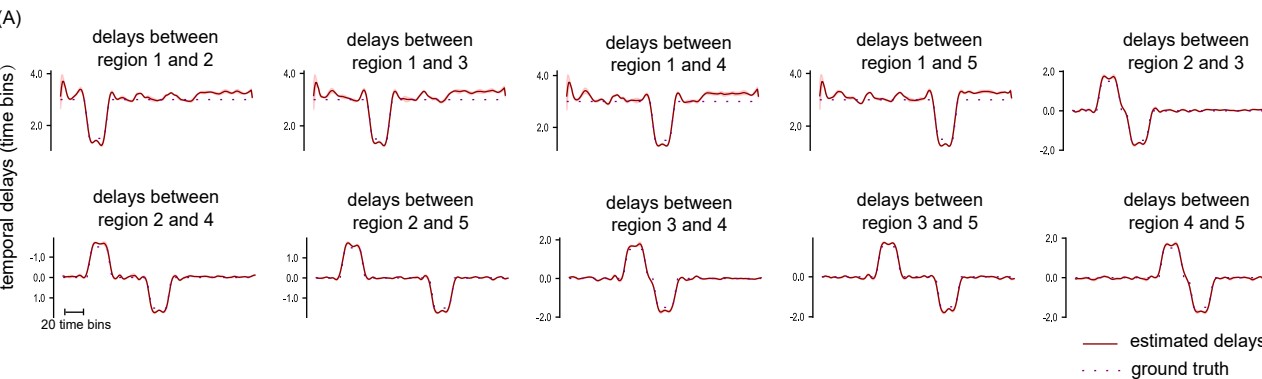

*Figure 7.* Estimated Pairwise Temporal Delays for Synthetic Data with Five Regions. Dashed lines indicate the ground truth delays, red lines show the estimated delays, and shaded areas represent the variance across different runs. The results demonstrate that the model accurately recovers the true delays. The increased MSE observed in Figure 8(A) is attributed to amplitude variability, which is not a practical concern, as the temporal delay patterns are well preserved.

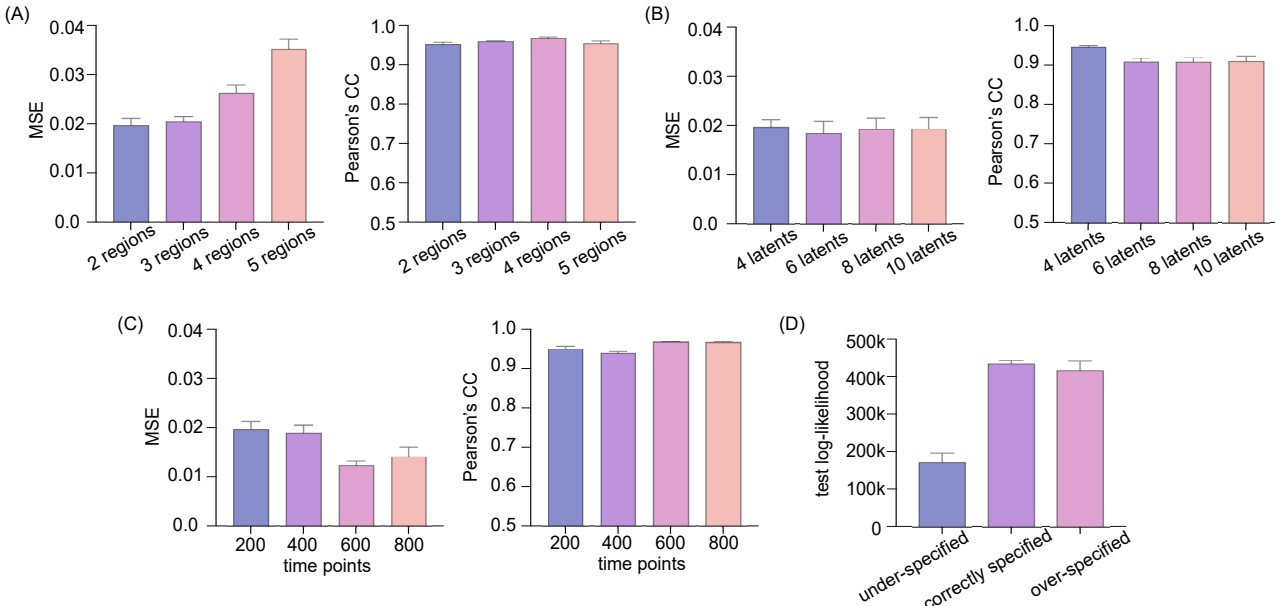

*Figure 8.* Additional Results for Synthetic Dataset. (A) Model evaluation using MSE and Pearson's correlation coefficient (CC) between estimated and ground truth delays across different numbers of regions. While MSE increases due to greater amplitude variability with more regions, CC remains stable, indicating reliable recovery of temporal delay patterns. See also Figure 7 for a visualization of estimated delays with five regions. (B) Model evaluation across varying numbers of latent variables, showing stable MSE and CC performance. (C) Model evaluation under different data lengths. Longer sequences yield lower MSE, likely due to improved estimation of underlying delays with more data. (D) Test log-likelihood comparison when the number of latent variables is under-specified, correctly specified, or over-specified. Both under- and over-specification result in lower log-likelihood and higher variance.

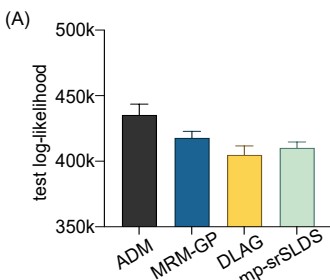

*Figure 9.* Test Log-Likelihood Comparison of ADM, DLAG, MRM-GP, and mp-srSLDS on the Synthetic Dataset used in Section 4.1. ADM consistently outperforms other methods by effectively capturing continuously time-varying temporal delays.

# H. Additional Results for Decoding Visual Stimuli in the Five-Region Brain Dataset

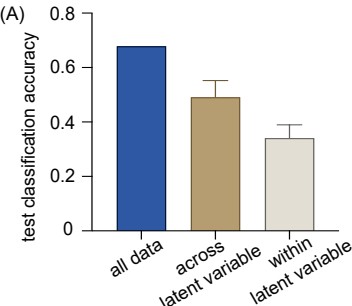

*Figure 10.* Decoding Visual Stimulus Orientation from the Visual Rostrolateral Area (VISrl). We analyze neural data from five regions and extract the learned latent variables. A linear decoder is then used to classify the orientation of visual stimuli ($0°$, $90°$, and $135°$) presented to the mouse during data collection. We evaluate decoding performance using three inputs: (1) raw observed neural activity from Visual Rostrolateral Area, (2) across-region latent variables (representing the communication subspace) of Visual Rostrolateral Area, and (3) within-region latent variables of Visual Rostrolateral Area. The results show that decoding directly from the observed neural data yields the highest test classification accuracy. Among the latent spaces, the communication subspace achieves higher accuracy than the within-region subspace. Notably, both the observed data and the communication subspace perform above random guessing, indicating that orientation information is preserved in the communication subspace of the Visual Rostrolateral Area, which is a region known to be involved in motion and spatial processing. Finally, the drop in accuracy from the observed data to the communication subspace is likely due to information loss from dimensionality reduction.

# I. Related Work on Neural Alignment and Task-Dependent Variability Modeling in Multi-Region Anslysis Interaction

Multi-region neural analyses increasingly rely on tools that can jointly characterise activity patterns within each area and the structured co-variability that links them. Williams et al. 2021 gives a geometric foundation for such comparisons by turning representational-similarity heuristics into proper metric spaces; they demonstrate that these "shape metrics" scale to surveys of 48 distinct visual areas in the Allen Brain Observatory, enabling distance-based clustering and embedding of whole-brain activity patterns. Safaie et al. 2023 pushes the multi-region perspective across species, showing that low-dimensional trajectories extracted from motor cortex recordings in both monkeys and mice align in a common latent space and remain predictive of movement even in the absence of overt behaviour, pointing to conserved circuit-level dynamics that transcend individual brains. Complementing these observational studies, Balzani et al., 2022 introduces TAME-GP, a probabilistic manifold model that factorises population variability into within-region and across-region components, imposes Gaussian-process priors for temporal smoothing, and infers single-trial latent dynamics that reveal task-dependent communication subspaces among multiple brain areas

