# OpenReview forum: "Learning Time-Varying Multi-Region Brain Communications  via Scalable Markovian Gaussian Processes"
_ICML.cc/2025/Conference — ICML 2025 oral_

### Official Review · Reviewer_sX24 · 2025-03-07

**Overall Recommendation:** 4

**Summary:**

This paper proposes a statistical model for estimating the time-varying delay of communication between multiple brain regions. This is achieved through low-dimensional latent variable modeling and incorporating multi-output Gaussian Process models. In addition to the modeling contribution, the paper exploits a connection between state-space models (SSM) and Gaussian Process (GP) with arbitrary temporally stationary kernels which allows the authors to develop fast and scalable inference models which scale logarithmically with the number of time points. Results are shown on a toy dataset as well as two neuroscience datasets with recordings from 2 and 5 regions respectively. The model discovers interesting interplay between feedforward and feedback flow of information across regions.

**Claims And Evidence:**

The claims regarding the speed of inference and the accuracy of the estimation for ground truth experiments are indeed supported by the results. For results on neuroscience datasets, the biological relevance of the estimated delays is still an open question. While the method provides valuable insights into the dynamic communication across regions, further validation experiments are required to support whether the delays have biological or causal relevance. Addressing the causality aspect of the estimated delays is beyond the scope of this paper and requires further careful investigation.

**Essential References Not Discussed:**

Nothing off the top of my mind.

**Experimental Designs Or Analyses:**

The datasets considered and the application of the model to those datasets is sounds and appropriate.

**Methods And Evaluation Criteria:**

Methods that jointly model the low-dimensional latent evolution of neural dynamics along with modeling the dynamic communication across regions did not exist before. Indeed this paper addresses an important and open question in neuroscience data analysis and the modeling framework used here is sensible and appropriate. In addition, the authors discover a deeper connection between SSM and GP with temporally stationary kernels with opens up new research direction to the broader ICML community.

**Other Comments Or Suggestions:**

**Minor issues**

- Please reword the following, there are two many communications in one sentence: [Understanding and constructing brain communications that capture dynamic communications across …].
- Please fix [Dicovering] -> discovering.
- Please fix [Dimmention] -> dimension.

**Other Strengths And Weaknesses:**

**Strengths**

- The paper is very well-formatted and well-written. I had a relatively easy job following the arguments and derivations.
- The accompanied code package follows standard code development practices.
- The connection between SSM and GP with temporally stationary kernels is of potential interest to the broader ML community.
- The method opens up new interesting analyses to the neuroscience community enabling new insights about multi-region communication.

**Areas of improvement**

- In some figures your description of the results has a causal flavor to it (feedforward vs. feedback direction of information). Imagine a scenario in which you have two regions exhibiting sine and cosine waves that are phase shifted; essentially it’s impossible to tell which one is sending/receiving information to the other, unless one region is manipulated and the effect is seen in the other region (but not in the opposite direction). Therefore it sounds like it’s impossible to claim anything causal from the estimated delay values. While this is a fundamental challenge associated with all the modeling approaches that do not consider causal manipulations, it’d be great if the authors can discuss this in the limitation section of their work. That said, I think there’s still a lot of value with the development of methods like the one proposed in the paper and it can help build insights and hypotheses that can then be causally tested.

- Both datasets presented in the paper use drifting grating stimuli which induce synchronous activity across brain regions; it’d be nice if the authors could show results on datasets with more complex sensory stimulus or datasets recorded from freely behaving animals.

**Questions For Authors:**

- Parallel scan $O(logT)$ that’s assuming you have access to O(T) GPUs, right? The arguments in the paper give the impression that $O(logT)$ comes for free. It'd be great if the authors could discuss this in the limitations section.
- Can you improve the diagonal covariance model and incorporate a full covariance model (or low-rank + diagonal)?
- I didn’t quite understand the inference, where are you computing the complete log likelihood? If you're running gradient descent isn't your algorithm essentially performing marginal likelihood optimization?
- Samples from a GP are nonlinear functions, looks like everything in your modeling framework is linear though, can you characterize the approximation gap you incur by treating a GP as an SSM?
- How do you deal with Poisson data? Can you expand the model to account for Poisson observations? In the current settings, do you apply any transformations to your spike counts to make them consumable by the model?
- Why not comparing against SLDS? Can you include these comparisons?
- What enforces orthogonality of the within vs. between variables? Also among “between” variables what makes them orthogonal? Is it not possible for all of the between variables to reflect the same temporal delay characteristics?
- You perform cross-validation on the delay, do you also perform cross-validation on the number of latents? If not, how do you set it? Can you show cross-validation results to get an overall sense of the sensitivity of the method to these parameters? Also can you include experiments where you set the number of latents to larger or smaller than the true value and see the effect? Specifically, if you set to larger this should result in some non-identifiability and inconsistency across runs or random initializations; this would specifically be an issue since you’re using the same number of between area latents which is practical assumption but might not be the the right assumption.
- How are the shaded errorbars computed? If you run the  method multiple times with different initializations, do you get the same result (same latents, same estimated delays)? How well defined is the problem? There seems to be many degrees of freedom in the model which suggests that multiple solutions can possibly co-exist (which is perhaps exacerbated by the existence of both positive and negative delays).
- Can you include state space visualizations too (e.g. PCA, or projections to the communication subspace)?
- Can you expand the simulation results to study the performance of the model (in terms of the recovery of the true delay parameter) as a function of number of regions, latents, delays, trials, smoothness of the kernel, etc.?
- What's the interplay between the choice of the kernel and the estimated delay. The estimated delay parameter seems to be dependent on the choice of kernel. In other words, if we change the kernel, the estimated delay parameter will be different too. Given this non-identifiability, how should we interpret the estimated delay values from a physical or biological standpoint?

**Relation To Broader Scientific Literature:**

The connection between SSM and GP can have potential applications that go beyond the results shown in the paper. In addition, the methodology is relevant to the computational and experimental neuroscience community and enables them to gain new insight into multi-region datasets.

**Theoretical Claims:**

The derivations seem correct to me. Apart from those the paper does not include theoretical claims. A few approximations are proposed in the paper. Addressing the quality of those approximations can be further investigated. I will explain this below in more detail.

---

> ### Author Rebuttal · Authors · 2025-03-31
>
> Dear Reviewer sX24,
>
> Thank you for the suggestion! We hope these improvements clarified your concerns, and that they can be taken into account when deciding the final score.
>
> The additional results are: (https://anonymous.4open.science/r/rebuttal-figures-for-ICML-2025-42E6/rebuttal_figures.pdf), including Rebuttal-Figures 1&4.
>
> > Claim anything causal from the delay?
>
> - We thank the reviewer for this thoughtful comment. We agree that delay or directional estimates from observational data alone do not prove causality. We do not make causal claims here; instead, we use these estimates to form hypotheses about across-region information flow. We acknowledge that only direct manipulations can establish true causal effects. Our model’s value is in highlighting putative, time-varying interactions, which can help pinpoint the circuits and time points most relevant for further causal testing. We will elaborate on this limitation in future revisions.
>
> > More complex sensory stimulus
>
> - Thanks for this suggestion. We agree that more complex stimuli offer exciting possibilities, and we see this as an important direction for future work.
>
> > Parallel scan access to O(T) GPUs.
>
> - Parallel scan algorithm relies on having enough parallel threads rather than literally requiring $T$ separate GPUs. We will clarify this in limitation part.
>
> > Improve the diagonal covariance model to a full covariance model?
>
> - In FA model, the latents approximates a full kernel GP, so the marginal covariance $\mathbf{C}Cov(\boldsymbol{x})\mathbf{C}^\top+\mathbf{R} $ is full.
>
> > Where are you computing the complete LL?
>
> - In the E-step, we use a Kalman filter/smoother to compute the posterior of the latents and compute the expected complete LL. In the M-step, we maximize this expectation with respect to model parameters using gradient descent, as part of the EM framework rather than directly optimizing the marginal LL. We will provide a pseudo algorithm in future revisions.
>
> > Approximation gap between SSM and GP.
>
> - Although our model is linear in state evolution, it can capture the nonlinear covariance structure of GP with a sufficient order $P$ in Eq. 5. In practice, choosing an appropriate $P$ results in performance that closely matches that of a GP.
>
> > How do you deal with Poisson data?
>
> - We preprocess using Gaussian smoothing, following prior work setting (Li et al., 2024). We acknowledge this is not ideal and developing a Poisson Kalmen Parallel scan is a key direction for future work.
>
> > Why not comparing against SLDS?
>
> - SLDS is not designed to capture temporal delays between regions. But we also show log-likelihood comparisons with multi-region SLDS in Rebuttal-Figure 4.
>
> > What enforces orthogonality of the within vs. across variables?
>
> - We do not claim strict orthogonality; instead, the FA model encourages decorrelation between variables through a block-diagonal observation matrix that assigns latents exclusively to shared or independent activity. Across-region variables use separate kernels, encouraging different delay characteristics; however, if the data indicates similar delays, the latents can reflect that similarity.
>
> > Do you also perform cross-validation on the number of latents?
>
> - Yes, we perform cross-validation on all hyperparameters, including the number of latents, in Figure 3(C). Also, in Rebuttal Figure 1(D), we test cases with larger and smaller latent numbers than true value. Incorrect latent settings result in larger variance across runs and lower test log-likelihood. We will address this issue in future revisions.
>
> > How are the shaded errorbars computed? Degrees of freedom?
>
> - The shaded error bars represent the variance of the learned delay across runs. We initialize the projection matrix $\mathbf{C}$ using CCA, which yields stable fits across random seeds. Our model incorporates a shared kernel length scale over time, which reduces the model’s degrees of freedom by determining the temporal dynamics and constraining the evolution of delays.
>
> > Can you include state space visualization?
>
> - We visualized our model’s communication subspace on both neural datasets in Figure 2(A) and Appendix E.
>
> > Performance vs number of regions, latents, and length?
>
> - We thank the reviewer for this insightful suggestion. As shown in Rebuttal Figure 1(A-C), the model demonstrates stable performance across different conditions.
>
> > The interplay between the kernel and the delay?
>
> - The estimated delay is an effective parameter that captures the time shift between signals, as defined by the chosen kernel. While different kernels may yield varying delay values, the directional information they convey remains consistent. The sign and relative ordering of delays are stable across reasonable kernel choices, as shown in prior work (Li et al., 2024), where a different kernel produced the same directional insights on the two regions data.
>
> Again, we thank the reviewer for insightful suggestions, which improved the quality of our paper!

---

> > ### Comment · Reviewer_sX24 · 2025-04-08
> >
> > Thank you for addressing the comments and including the new results, I have adjusted my score accordingly.

---

> > > ### Author Response · Authors · 2025-04-08
> > >
> > > Thank you very much for updating the score! We sincerely appreciate your insightful feedback.

---

### Official Review · Reviewer_MsHr · 2025-03-14

**Overall Recommendation:** 4

**Summary:**

This paper extends GP-based methods for modeling multi-region neural recordings to the case where temporal delays in communication can dynamically shift. This is achieved by a novel combination of GPs with state-space models. The authors evalute their work on both synthetic and real multi-region neural recording datasets.

**Claims And Evidence:**

Overall, the paper did a good job supporting its claims with a thorough and extensive evaluation. They are able to reconstruct the true time-varying delays in the setting of synthetic data. And they show that they can fit real-world data, and the results make sense in the context of known details about the dynamics of the studied regions.

**Essential References Not Discussed:**

There are no essential references missing from this paper as far as I'm aware of the literature.

**Experimental Designs Or Analyses:**

I looked over the experimental design of the main results and found them to be sound. The synthetic data with known time-varying temporal delays is a great place to start as a sanity check. And the real-world datasets show the scalability of the method to more than two regions. It also demonstrates that it fits to real data, and illustrates the type of hypotheses it can help generate for further testing. It would be interesting to see how well the model reconstructs true delays in the setting where there are more than two regions though. This could be tested with a synthetic model similar to the proposed one, but with more than two regions. Additionally, the results show that the model is fairly consistent across seeds, which is a nice property for a modeling framework.

**Methods And Evaluation Criteria:**

The proposed methods and evaluation criteria (both benchmarks and metrics) make sense for this problem setting. The real-world data analysis is more of a proof-of-concept, which is appropriate for the venue.

**Other Comments Or Suggestions:**

No additional comments.

**Other Strengths And Weaknesses:**

While not the primary concern of this paper, it is unclear how well this framework can account for the ill-posedness of inferring time-varying messages with static delays versus time-varying delays. It's possible that some of what appears to be time-varying in the delay of data is more due to the time-varying nature of the messages than something about the actual communication channel. But this comes down to the interpretation of the model, which is an important but different problem than the focus of the paper.

**Questions For Authors:**

- How well does the model reconstruct true time-varying delays in data generated by a synthetic model with more than two regions?
- To what degree is the inference of delays vs. communications an ill-posed problem? Is there a degeneracy where the true message could have static delays in terms of the transmission channel but it looks like a time-varying delay due to the time-varying nature of the communications?

**Relation To Broader Scientific Literature:**

This paper extends state-of-the-art GP-based multi-region brain modeling techniques which focus on modeling inter-region delays in communication. Prior work primarily focuses on static delays, but this method considers the case where they are influenced by cognitive states. Their developed method also combines GPs with state-space models in an interesting way which may be of indepdent interest outside of neuroscience.

**Theoretical Claims:**

I looked over the derivations in Appendix A and nothing popped out to me as incorrect.

---

> ### Author Rebuttal · Authors · 2025-03-31
>
> Dear Reviewer MsHr,
>
> Thank you for the encouraging feedback and suggestion! We have made several clarifications according to your questions and comments. We hope these sufficiently clarified your concerns, and that they can be taken into account when deciding the final review score.
>
> The additional results are provided (https://anonymous.4open.science/r/rebuttal-figures-for-ICML-2025-42E6/rebuttal_figures.pdf), including:
>
> - Delay estimation on the synthetic dataset with five regions (Rebuttal-Figure 2).
>
> > It would be interesting to see how well the model reconstructs true delays in the setting where there are more than two regions though.
>
> - We thank the reviewer for this insightful suggestion. In Rebuttal-Figure 2, we present the estimated time-varying delays from synthetic data with five regions. Rebuttal-Figure 1(A-C) further shows model performance in terms of MSE and Pearson’s correlation coefficient (CC) between the estimated and ground truth delays as the number of regions increases. While MSE rises due to greater amplitude variability, CC remains stable, indicating reliable recovery of the temporal delay patterns. As shown in Rebuttal-Figure 2, this variability in amplitude is not a practical concern, as the temporal patterns, which are crucial for understanding brain communication, are well recovered. We will add these results in the revised version.
>
> > To what degree is the inference of delays vs. communications an ill-posed problem? Is there a degeneracy where the true message could have static delays in terms of the transmission channel but it looks like a time-varying delay due to the time-varying nature of the communications?
>
> - We thank the reviewer for raising this question. Separating the effects of time-varying delays from changes in the message content is inherently challenging. In principle, the inference problem is ill-posed because the same observed data could be explained either by static delays combined with time-varying messages or by genuinely time-varying delays. In other words, there is a potential degeneracy: even if the true communication channel has static delays, the variability in the messages may cause the inference process to interpret these as time-varying delays.
>
> - To mitigate this issue, our model incorporates a regularization strategy by using a shared length scale parameter $l$ across all time points. As described in Sec. 2.2, the state transition matrix $\mathbf{\hat{A}}_t$ is uniquely determined by the delay $\theta_t$ and the length scale $l$. By sharing $l$ across time, we constrain the temporal evolution of the delay parameters, enforcing similar temporal dynamics and reducing the risk of misattributing variability in the messages to changes in delays. Furthermore, our synthetic experiments with known ground truth demonstrate that the model reliably recovers the true dynamic delays, indicating that the inferred time-varying delays reflect genuine changes in the communication pathways rather than artifacts arising from the message content.
>
> - However, we acknowledge that further research could explore additional constraints or independent measures to better address this problem, but our current results suggest that the degeneracy is minimal in the context of our application.
>
> Again, we thank the reviewer for the comments, constructive feedback, and insightful suggestions, which improved the quality of our paper. If you have further questions, we are happy to discuss them!

---

> > ### Comment · Reviewer_MsHr · 2025-04-03
> >
> > I thank the authors for providing additional experiments showing reconstruction of known true delays in synthetic data! And for considering my concerns about the degeneracy of true messages with static versus time-varying delays.

---

> > > ### Author Response · Authors · 2025-04-03
> > >
> > > We again thank the reviewer for your encouraging feedback and for appreciating our additional experiments!

---

### Official Review · Reviewer_9Ksf · 2025-03-14

**Overall Recommendation:** 4

**Summary:**

Modeling neural activity across networks of populations of neurons across multiple brain regions is critical to understand neural computation and how information is processed. While the recent recoding technological advances have made it possible to acquire the data, modeling tools are limited in their ability to capture all the variability present in it. To address some of these limitations, the authors present a new model that extends latent space models to capture time varying variability across brain regions while allowing for temporal delays between them. The authors also introduced an inference procedure to reduce computational complexity. They test the model in a synthetic dataset and two neural datasets showing the feasibility of the approach.

**Claims And Evidence:**

The authors clearly frame their work and provide theoretical proof and empirical results to back them. The authors present results for one synthetic dataset and two neural datasets, showing the feasibility of their model. However, the authors motivate their work by stating that existing methods fail to capture neural representations that could provide additional insight into neural processing, but they fail to present novel neuroscience discovery beyond corroboration with results surfaced by existing models. Comparatively showing how this model can improve interpretability or drive new insight is critical to understand the impact of the contributed work.

**Essential References Not Discussed:**

The aforementioned references should be added to illustrate alternative alignment methods for neural recordings without explicit modeling of temporal dynamics (Williams et al NeurIPS 2021, Safaie et al. Nature 2023) or that capture task and behavioral information (Balzani et al. ICLR 2023).

**Experimental Designs Or Analyses:**

As mentioned previously, the design and analysis address the feasibility of the approach but lacks to fully illustrate the advantages of the model with respect to alternative models or the ability to drive new scientific insight. Including comparisons to simpler methods, which are expected to fail to capture the neural variability, and a comparison beyond log likelihood estimates would further strengthen the presented results. Possible extensions would be to include comparisons of the latent representations across models, or decoding performance with respect to relevant neural computation variables, such as visual stimulus class.

**Methods And Evaluation Criteria:**

The methods are well motivated and described. The authors compared their model to alternative existing models that explicitly model temporal dynamics and delays, but do not provide systematic comparisons to other static methods such as vanilla Procrustes alignment (Williams et al NeurIPS 2021, Safaie et al. Nature 2023). Including such comparisons would be critical to evaluate the tradeoffs between computational cost, data demands and explanation capabilities across models.
They minimally evaluate the proposed model in a synthetic dataset, but the authors fail to show comparative performance with other models in this data. Moreover, they could have further used this setting to assess the applicability and limitations of the model (e.g. how does the model behave with respect to the latent dimensionality, or number of areas, or length of recording?). The authors present results on two sets of neural data, where they prove that their method is more computationally efficient and provides efficient reconstruction of the neural responses and captures the temporal delays between regions. It would be relevant to assess not only the delays and directionality of the connections between regions, but also i) the neural representations in the communication space and ii) the task relevant information present between the communications subspaces. This would further highlight the potential of the model to drive neuroscientific discovery.

**Other Comments Or Suggestions:**

Axes ticks and labels are missing in multiple figures, including Fig. 1A, 4 and 5 both x- and y- axes, Fig1C, 2A,D , 3A, 6 x-axis.

**Other Strengths And Weaknesses:**

The method uses vanilla FA to estimate the latent dimensionality, and it is not clearly listed as a limitation or design choice. Why not use likelihood estimates from the model itself?

**Questions For Authors:**

Does the model support different dimensionality across different latent spaces?

**Relation To Broader Scientific Literature:**

The authors adequately motivate their work and present alternative methods. However, they could include references to other simpler neural alignment methods (Williams et al NeurIPS 2021, Safaie et al. Nature 2023). And add a references to methods that capture communication spaces between brain regions and also model the task variability, such as Balzani et al. ICLR 2023.

**Theoretical Claims:**

The theoretical claims are correct, and full proofs are shown in the supplementary material.

---

> ### Author Rebuttal · Authors · 2025-03-30
>
> Dear Reviewer 9Ksf,
>
> Thank you for the constructive comments! We have made several improvement according to your questions and comments. We hope these sufficiently clarified your concerns, and that they can be taken into account when deciding the final review score.
>
> The additional results are provided (https://anonymous.4open.science/r/rebuttal-figures-for-ICML-2025-42E6/rebuttal_figures.pdf), including:
>
> - A comparison of our model with existing methods on the synthetic dataset (Rebuttal-Figure 4(C)).
> - The model performance vs. the number of regions, latents, and length on the synthetic dataset (Rebuttal-Figure 1(A-C)).
> - Decoding visual stimulus-related information from the learned latents (Rebuttal-Figure 3).
>
> > The aforementioned references should be added (Williams et al. NeurIPS 2021, Safaie et al. Nature 2023, Balzani et al. ICLR 2023).
>
> - Williams et al. (NeurIPS 2021) introduce a metric framework for comparing static neural representations, Safaie et al. (Nature 2023) show preserved latent dynamics across subjects using alignment, and Balzani et al. (ICLR 2023) propose a task-aligned model capturing within- and across-region neural variability. In contrast, our work focuses on modeling delay-based brain communication across regions, aiming to capture dynamic, time-varying delays that mediate across-region interactions. This enables insights into the directionality and strength of functional coupling.
>
> - Therefore, we respectfully disagree that our work should be directly compared to alignment-based approaches such as Williams et al. (NeurIPS 2021) and Safaie et al. (Nature 2023), as our modeling goals differ. On the other hand, we greatly appreciate the novel perspective introduced by Balzani et al. (ICLR 2023) on task alignment within communication subspaces. We will include a discussion of their work in our revised manuscript to clarify how it complements and contrasts with our approach.
>
> > They minimally evaluate the proposed model in a synthetic dataset, but the authors fail to show comparative performance with other models in this data.
>
> - We thank the reviewer for raising this concern. In Rebuttal-Figure 4(C), we present the test observation log-likelihoods on the time-varying synthetic dataset used in Sec. 4.1. Our model outperforms existing multi-region modeling methods, demonstrating its effectiveness in capturing time-varying multi-region communications.
>
> > How does the model behave with respect to the latent dimensionality, or number of areas, or length of recording?.
>
> - We thank the reviewer for this valuable suggestion. In Rebuttal-Figure 1(A-C), we evaluate our model using MSE and Pearson's correlation coefficient (CC) between estimated and ground truth delays under varying numbers of regions, latent dimensions, and lengths. The model shows stable performance across conditions. While MSE increases due to amplitude variability with more regions, CC remains stable, indicating reliable recovery of temporal delay patterns. As shown in Rebuttal-Figure 2, this amplitude variability is not a practical concern because the temporal patterns, which are crucial for understanding brain communication, are well preserved.
>
> > The method uses vanilla FA to estimate the latent dimensionality, ..., why not use likelihood estimates from the model itself?
>
> - We follow the strategy from prior work (Gokcen et al., 2022), see Sec. 4.3, using FA to estimate total latent dimensionality and then performing a grid search with model log-likelihood to determine across- and within-region dimensions. This approach balances accuracy with computational efficiency, as a full grid search of latent dimensionality is time-consuming.
>
> > Axes ticks and labels are missing in multiple figures.
>
> - We apologize for the oversight and thank the reviewer for pointing it out. We will add axis ticks and labels in the revised version.
>
> > Possible extensions would be to include comparisons of the latent representations across models, or decoding performance with respect to relevant neural computation variables, such as visual stimulus class.
>
> - Thank you for the suggestion. We have visualized our model’s latent representations on both neural datasets in Figure 2(A) and Appendix E. In the revised version, we will include latent representations from other models for comparison. Additionally, Rebuttal-Figure 3 demonstrates that decoding from our latent variables shows stronger task-related information (e.g., grating orientations) in the communication subspaces compared to the within-region subspaces. We also view the discovery of further task-relevant information in these subspaces as a promising direction for future work.
>
> > Different dimensionality across different latent spaces?
>
> - Yes, we allow the number of within-region latents to vary over regions.
>
> Again, we thank the reviewer for the insightful suggestions, which improved the quality of our paper. If you have further questions, we are happy to discuss them!

---

> > ### Comment · Reviewer_9Ksf · 2025-04-06
> >
> > I thank the authors for the detail comments and additional experiments. I adjusted the score accordingly. Please add the new results and discussed reference points to the final manuscript.

---

> > > ### Author Response · Authors · 2025-04-06
> > >
> > > Thanks very much for your thoughtful feedback and for reconsidering your score. We sincerely appreciate your time and effort in reviewing our work. We will make sure to incorporate the new results and discussed reference points into the final paper.

---

### Official Review · Reviewer_Wfpo · 2025-03-17

**Overall Recommendation:** 4

**Summary:**

This submission describes an approach for inferring latent factors underlying shared neural responses across brain areas. Notably, the approach enables inferring a continuous and time-varying delay factor that captures the temporal delay between two brain areas. The submission formulates this as both a Gaussian Process model and state-space model (SSM) and leverages the SSM setup to enable faster inference via parallel scans. The method is validated in a simulated dataset and demonstrated on two experimental applications for the analysis of visual data. Importantly, the method found time-varying delay responses and was applied to more than two brain areas.

**Claims And Evidence:**

The submission presents convincing evidence that the approach enables inference of time-varying delays in brain communication.

**Essential References Not Discussed:**

N/A

**Experimental Designs Or Analyses:**

Yes, I checked the details for the simulated and two experimental data analyses.

**Methods And Evaluation Criteria:**

Yes, the datasets and methods make sense. The submission evaluates the method using standard approaches on both simulated datasets and two relevant neuroscience datasets.

**Other Comments Or Suggestions:**

N/A

**Other Strengths And Weaknesses:**

Strength:

The submission nicely demonstrated the generality of their proposed approach, showing how under various GP kernels they could relatively accurately approximate a GP regression using their SSM approximation.

Weakness:

Figure 3(B) is challenging to understand. I suggest presenting these results in a different format, potentially via a matrix where each element represents a signed pairwise connection.

**Questions For Authors:**

Can the authors provide additional details on the formulation and fitting of the time-varying delay? It appears to be one of the crucial innovations. I'm curious on how flexible this model is and whether it presents any issues in fitting. The paper states that the resulting $\hat{A}_t$ are still constrained at each time point via a shared length scale. However, it is not obvious to me how strongly that constrains these parameters.

Additionally, Appendix B does not appear to describe the time-varying dynamics, as $A$ is not indexed by $t$ in the EM equations. Is there a different objective for the time-varying formulation? If so, it would be very helpful and important to put that in the appendix as well.

**Relation To Broader Scientific Literature:**

This submission is related to a line of important work developing methods for the analysis of neural recordings that span multiple brain areas. In particular, this submission builds on previous work on Gaussian process models for capturing shared and private variability across brain areas, potentially with time delays.

**Theoretical Claims:**

N/A

---

> ### Author Rebuttal · Authors · 2025-03-30
>
> Dear Reviewer Wfpo,
>
> Thank you for the encouraging feedback and practical suggestions! We have made several clarifications according to your questions and comments. Hopefully these will resolve most of your concerns, and that they can be taken into account when deciding the final review score.
>
> > Figure 3(B) is challenging to understand. I suggest presenting these results in a different format, potentially via a matrix where each element represents a signed pairwise connection.
>
> - We thank the reviewer for pointing this out. Figure 3(B) is intended to highlight the time-varying nature of temporal delays across five brain regions. It visualizes the learned temporal delays from Figure 3(A) at time points $t=3$ and $t=50$. In the figure, the orange arrows indicate the direction of communication: a positive delay denotes forward communication, while a negative delay indicates feedback communication. Additionally, the length of each edge represents the absolute value of the corresponding temporal delay.
>
> - To provide a clearer and more quantitative view, we will add a matrix alongside the network diagram to explicitly show both the magnitude and sign of the learned temporal delays.
>
> > Can the authors provide additional details on the formulation and fitting of the time-varying delay? It appears to be one of the crucial innovations. I'm curious on how flexible this model is and whether it presents any issues in fitting. The paper states that the resulting $\mathbf{\hat{A}}_t$ are still constrained at each time point via a shared length scale. However, it is not obvious to me how strongly that constrains these parameters.
>
> - At each time step $t$, we construct a time-specific Markovian GP conditioned on the MOSE kernel corresponding to that time step. The state transition matrix $\mathbf{\hat{A}}_t$ is uniquely determined by the delay $\theta_t$ and length scale $l$ as mentioned in Sec. 2.2.  While the delay $\theta_t$ is allowed to change over time, the length scale $l$, which is shared across all time steps, limits the freedom of $\mathbf{\hat{A}}_t$. This shared length scale fully determines the temporal dynamics of the latent variables and thus constrains the evolution of dynamics from $\mathbf{\hat{A}}_t$ to follow the same smoothness. In practice, this allows the model to flexibly capture time-varying delays while maintaining similar temporal dynamics across time.
>
> - The blue and red latent variables shown in Figure 1(A) illustrate this behavior: they share overall temporal dynamics but exhibit different delays at each time point. Empirical results in Sec. 4.1 further confirm that this shared length scale leads to robust and smooth delay estimates without fitting issues if a reasonable initialization is used, e.g., setting a relatively large length scale to encourage initial smoothness. This initialization aligns with standard practice in GP regression, where initializing with a larger length scale helps guide optimization during early iterations.
>
> > Additionally, Appendix B does not appear to describe the time-varying dynamics, as $\mathbf{A}$ is not indexed by in the EM equations. Is there a different objective for the time-varying formulation? If so, it would be very helpful and important to put that in the appendix as well.
>
> - We apologize for the confusion regarding Appendix B. In our time-varying extension, the underlying EM objective remains the same: maximizing the expected complete-data log-likelihood, but the state-space model is augmented to include time-specific transition matrices $\mathbf{\hat{A}}_t$ and process noise covariances $\mathbf{\hat{Q}}_t$. The derivations for the EM updates are analogous to those in Appendix B, with the key difference that the E-step now involves running a vectorized Kalman filter and smoother over the entire time sequence. As a result, the shapes of $\mathbf{\hat{A}}$ and $\mathbf{\hat{Q}}$ become $(T, NP, NP)$. We will revise Appendix B to explicitly incorporate these modifications and clearly describe the time-varying dynamics.
>
> Again, we thank the reviewer for the comments, constructive feedback, and insightful suggestions, which significantly improved the quality of our paper. If you have further questions, we are happy to discuss them!

---

> > ### Comment · Reviewer_Wfpo · 2025-04-09
> >
> > Thank you for the thorough response. I have increased my score to Accept in light of these improvements.

---

> > > ### Author Response · Authors · 2025-04-09
> > >
> > > We thank the reviewer for the encouraging feedback and updating the score!

---

### Decision · Program_Chairs · 2025-05-01

**Decision:**

Accept (oral)

**Comment:**

This paper addresses itself to the estimation of communication between different recording sites in multi-region brain activity data. This problem has received increasing attention in the literature as more and more experiments generate these data, and as a result, several methods have been proposed, all of which necessarily trade off computational efficiency and flexibility.

This work proposes to approximate a flexible but inefficient Gaussian Process model with between-region time delays by a multi-lag state space model (SSM). While similar papers such as Li et al. (2024) have attempted this link, this paper provides an alternative approach that allows for both a) inferring time-varying delays between regions and b) computationally efficient parallel scan operations.

Reviewers found the paper clearly written and convincing, with well-chosen benchmarks and model comparisons. While the model does make some approximations (detailed by the authors), it clearly represents a new state of the art in analyzing communication between recording sites in multi-region brain data.